# NFPO: Stabilized Policy Optimization of Normalizing Flow for Robotic Policy Learning

## Abstract

Deep Reinforcement Learning (DRL) has experienced significant advancements in recent years and has been widely used in many fields. In DRL-based robotic policy learning, however, current *de facto* policy parameterization is still multivariate Gaussian (with diagonal covariance matrix), which lacks the ability to model multi-modal distribution. In this work, we explore the adoption of a modern network architecture, i.e. Normalizing Flow (NF) as the policy parameterization for its ability of multi-modal modeling, closed form of log probability and low computation and memory overhead. However, naively training NF in online Reinforcement Learning (RL) usually leads to training instability. We provide a detailed analysis for this phenomenon and successfully address it via simple but effective technique. With extensive experiments in multiple simulation environments, we show our method, NFPO could obtain robust and strong performance in widely used robotic learning tasks and successfully transfer into real-world robots.

## 1 Introduction

Deep Reinforcement Learning (DRL), as a machine learning field studying sequential decision making, has received tremendous research interest and advancements in recent years (Mnih et al., 2015; Silver et al., 2016; Ouyang et al., 2022; DeepSeek-AI et al., 2025). With reduced Sim-to-Real gap (Margolis et al., 2022b; Fu et al., 2022), nowadays the policies trained in parallel simulation environments by RL could be transferred with unprecedented easiness to real world robotic systems. This avoids the need to manually collect real data which is highly expensive and cumbersome, significantly speeding up the researches and developments. Through this way, many policies trained by DRL have been successfully deployed and enabled impressive accomplishments in generating smooth, autonomous motions on real-world bipedal, quadruped and humanoid robots (Li et al., 2023; Lee et al., 2020; Ze et al., 2025; Margolis et al., 2022a; Radosavovic et al., 2024; Zhang et al., 2025; Shao et al., 2025).

However, current *de facto* policy parameterization in DRL robotic policy learning is still multivariate Gaussian with diagonal covariance matrix which is known to have poor ability to model mutli-modal distributions. This is in stark contrast to another paradigm, Supervised-Learning-Based robotic policy learning (a.k.a Behavior Cloning where a policy is trained to mimic pre-collected behavior dataset) where modern policy networks like Diffusion Policy (Chi et al., 2023; Liao et al., 2025) and Transformers (Brohan et al., 2023b; **?**;a) are in wide adoption.

While many recent works have studied the integration of diffusion-based policies into Online Reinforcement Learning paradigm, most of them are built on off-policy methods (like Soft Actor Critic (SAC) (Haarnoja et al., 2018)) for stronger sample efficiency. While in robotic learning, simulation environments offer massive samples with *trivial cost* and it's 1) computation efficiency, 2) memory consumption and 3) robustness towards multiple simulator and reward function settings are of more importance. Under this setting, on-policy methods like Proximal Policy Optimization (PPO) (Schulman et al., 2017a;b) are more favored and have delivered countless successes. Unfortunately, it remains unclear how to integrate modern multi-modal-modeling networks into Policy Optimization's paradigm, and train control policy purely from scratch[1]. A table comparison between our method and related works could be found in Table 1.

---

[1]There also exist methods that perform purely offline or offline-to-online finetuning of diffusion or Transformer policies via RL for robots. But we focus on from-scratch DRL learning where no dataset is present.

Table 1: Comparison of NFPO to related methods. Detailed explanations of how we choose ✓or ✗ and the references are provided in Appendix A.7.

| Algorithm | Multi-modality | On-policy | Memory & Computation | Real World Deployment |
|---|---|---|---|---|
| FastTD3 | ✗ | ✗ | ✗ | ✓ |
| Meow | ✓ | ✗ | ✓ | ✗ |
| MaxEntDP | ✓ | ✗ | ✗ | ✗ |
| GenPO | ✓ | ✓ | ✗ | ✗ |
| FPO | ✓ | ✓ | ✓ | ✗ |
| **Ours (NFPO)** | ✓ | ✓ | ✓ | ✓ |

In this work, we aim to bridge this gap by designing a new method which is 1) computation and memory efficient, 2) robust towards multiple simulator and reward function settings and 3) simple with few code changes to current training pipeline. We choose Normalizing Flow (NF) as it naturally fits all above requirements. However, naively combining NF with Policy Optimization would cause severe training and numerical instability. We provide detailed analysis and show how to address it with simple but effective techniques. In summary, our contributions are:

1. Aiming at robotic multi-modal policy learning, we integrate NF into PPO, analyze the reasons of its training instability and propose methods to address it.

2. We extensively test our method in multiple widely-used simulation environments (IsaacGym, Mujoco-playground and IsaacLab) and demonstrate our method (with the same set of configurations) could obtain competitive performance compared to state-of-the-art Gaussian PPO implementation.

3. We successfully transfer the policy trained with NFPO to real-world robots to show it could perform various tasks like locomotion and motion tracking.

## 2  RELATED WORKS

**Deep Reinforcement Learning**. DRL has experienced tremendous advancements in recent years. From on-policy methods like TRPO (Schulman et al., 2017a), PPO (Schulman et al., 2017b) to off-policy methods like DQN (Mnih et al., 2015), DDQN (van Hasselt et al., 2015), TD3 (Fujimoto et al., 2018) and SAC (Haarnoja et al., 2018), a main research direction is to improve the *sample-efficiency* which measures how many samples a method needs to achieve certain return, as humans could learn very efficiently with few interactions. By leveraging learnable dynamics and reward models, Model-based RL like Dreamer (Hafner et al., 2025) and TDMPC (Hansen et al., 2024) further increases the sample-efficiency compared to their model-free alternatives. Inspired by recent innovations in neural network architecture like Diffusion Model, Transformers and Normalizing Flows, another line of works targeting on combining these powerful networks into DRL's framework has also emerged as in Chao et al. (2024); Ding et al. (2025); Dong et al. (2025). In this work, we try to integrate NFs into on-policy PPO's pipeline as the latter is widely used in robotic policy learning.

**Normalizing Flow**. Normalizing Flow is a kind of generative models, featured by bijective mapping and closed calculation of log probability, compared to other generative methods like GAN (Goodfellow et al., 2014), Diffusion Models (Ho et al., 2020) and Score-based Models (Song et al., 2020). Since the early works like Dinh et al. (2015; 2017), NFs have also gained significant improvement. Especially in Zhai et al. (2025), the flexibility and generation quality of NFs have gained much improvement by using attention techniques. In DRL, the efficient and accurate calculation of log probability has made NFs an appealing policy parameterization and some works have explored the combination of NFs and DRL as in (Chao et al., 2024; Ghugare & Eysenbach, 2025). However, to the best of our knowledge, we are the first to integrate NFs into on-policy RL setting.

**Robotic Policy Learning**. Thanks to the rapid advancements of GPU-based parallel simulation environments and reduced Sim-to-Real gap (Kumar et al., 2021; Fu et al., 2022), Robotic Policy Learning has garnered great progress. From bipedal, quadruped robots with locomotion skills like parkour (Miki et al., 2022), fast running (Margolis et al., 2022c) to humanoid robots where teleoperation (Ze et al., 2025; He et al., 2024) and motion tracking (Liao et al., 2025) have enabled dancing,

kicking and somersault, many agile and stable behaviors have been learned by DRL pipeline. Alternatively, in Supervised-Learning-based robotic policy learning, modern network architecture like Transformers and Diffusion Models have shown great advantages for their expressiveness, like in Chi et al. (2023); Ren et al. (2024); Brohan et al. (2023a); Octo Model Team et al. (2024). In this work, we try to explore the integration of NFs as policy parameterizations in DRL paradigm.

## 3  BACKGROUND

**Reinforcement Learning**. Reinforcement Learning (RL) aims to solve sequential decision-making problem which is formulated as a Markovian Decision Process (MDP): $\mathcal{M} = \{\mathcal{S}, \mathcal{A}, \mathcal{R}, P, \gamma, \mathcal{S}_0\}$ where $\mathcal{S}$ is the set of all *states*, $\mathcal{A}$ is the set of all *actions*, $\mathcal{R} : (s_t, a_t) \to \mathbb{R}$ is the reward function that gives a scalar value for given state action pair, $P : s_{t+1} \sim P(s_t, a_t)$ is the transition model that gives the next state given current state action, $\gamma$ is the discount factor and $\mathcal{S}_0$ is the distribution of initial states. RL is to find a policy function $\pi$ that maximizes the expected return in $\mathcal{M}$:

$$\pi = \arg\max_{\pi \sim \Pi} \mathbb{E}\left[\sum_{t=0}^{\infty} \gamma^t \mathcal{R}(s_t, a_t)\right] \tag{1}$$

**Proximal Policy Optimization**. As a notable variant of on-policy RL, Proximal Policy Optimization (PPO) (Schulman et al., 2017b) performs policy optimization via advantage-based clip updates:

$$\pi_{\text{new}} = \arg\max_{\theta} \mathbb{E}_{a \sim \pi_{\text{old}}, s \sim P(s,a)}\left[\min\left(r(\theta)\tilde{A}(s,a), \text{clip}(r(\theta), 1-\epsilon, 1+\epsilon)\tilde{A}(s,a)\right)\right] \tag{2}$$

where $r(\theta) = \frac{\pi_\theta(a|s)}{\pi_{\text{old}}(a|s)}$ is the ratio of action likelihood and $\tilde{A}(s,a)$ is the estimated advantages (e.g., Generalized Advantage Estimation (GAE) as in Schulman et al. (2018)). As it requires log probability of action given state, the common policy parameterization is Gaussian with learnable mean and diagonal covariance matrix: $\pi(s) = \mathcal{N}(\mu_\theta(s), \text{diag}(\sigma_\theta(s)))$.

**Normalizing Flow**. Normalizing Flow (NF) is a bijective mapping with learnable components $f_\theta : \mathbb{R}^D \to \mathbb{R}^D$ between data distribution $p(x)$ and prior distribution $q(z)$. It's designed such that the inverse $f_\theta^{-1}$ and the determinant of its Jacobian $|\frac{df_\theta(x)}{dx}|$ is in closed form and could be efficiently computed. The prior distribution is chosen as simple ones like Normal distribution. Then the data density could be expressed as:

$$p(x) = q(f_\theta(x))\left|\frac{df_\theta(x)}{dx}\right| \tag{3}$$

Then we could use Maximum Likelihood Estimation (MLE) based training objectives $\theta = \arg\max_\theta \mathbb{E}_x[\log q(f_\theta(x)) + \log|\frac{df_\theta(x)}{dx}|]$, and perform *sampling* via $z \sim q(z); x = f_\theta^{-1}(z)$ and *inference* via $x \sim p(x); z = f_\theta(x)$.

**RealNVP**. An important NF variant is RealNVP (Dinh et al., 2017) with *coupling layers*:

$$f_\theta(x)_d = x_d \tag{4}$$
$$f_\theta(x)_{\backslash d} = x_{\backslash d} \odot \exp(s_\theta(x_d)) + t_\theta(x_d) \tag{5}$$

where $d$ is a set containing certain indexes, $\backslash d \left(= \{i | 1 \le i \le D, i \in \mathbb{N}^+\}\backslash d\right)$ is a set containing remaining indexes, $\odot$ is element-wise production and $s_\theta$, $t_\theta$ are 2 neural network from $\mathbb{R}^{|d|}$ to $\mathbb{R}^{|D-d|}$. From above definition, the Jacobian is: $|\frac{df_\theta(x)}{dx}| = \exp\left[\sum s(x_d)\right]$. Following the original work, we employ alternating odd-even strategy to partition the index set in which 3 or more stacked layers is needed to allow each dimension in $x$ to influence each other.

## 4  METHOD

With above definition of PPO and NFs, we could build an optimization pipeline where NFs are used as policy parameterizations instead of Gaussian. Thanks to its closed form of log probability, the changes mainly involve a new calculation of $\pi(a|s)$ and sampling from prior distribution to generate action samples. All other components and formulas could be reused. However, a naive applying of above methods would likely result in training instability, for the following reasons:

1. **Overfitting.** Expressive multi-modality policies are prone to overfitting than Gaussian-based counterparts. As the modeling ability increases, the network could find a cheap optimization strategy that assigns high log-probability to all points with positive advantages and vice versa. An illustrative diagram is in Figure 1.

2. **Exponential values.** the $\exp(s_\theta(x_d))$ used in RealNVP applies exponential transformations on the output of neural networks. This further increases the tendency of overfitting as simply increase $s_\theta(x_d)$ would increase the training objective significantly.

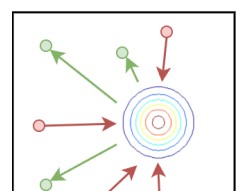
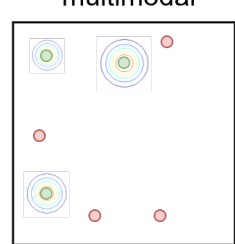

Figure 1: Illustrative diagram of multi-modal overfitting. Green samples are those whose probability needs increase while red samples need decrease. In left figure, the unimodal model cannot fit all these training signals but in the right figure a multimodal model overfits.

3. **Unbounded Output.** Unlike Gaussian whose log-probability is roughly bounded (so long as the standard deviation is not infinitesimal), neural network's output could be unbounded and lead to numerical instability values.

To further illustrate above phenomenon, we perform an experiment using UnitreeRLGym's g1 environment. Specifically, we build a 4-layered RealNVP as policy parameterization and integrate it into PPO's training pipeline, the training result is in Figure 2 under name of s_none.

From the result, we find the performance of s_none increases in the first and middle stage of training. However, the determinant of its Jacobian keeps increasing to very large values then it triggers numeric instability and training crashes.

**Solution.** As observed above, the training instability of NFs under Policy Optimization is mainly caused by the exponential transformation of unbounded output of $s_\theta(x)$. A simple yet effective technique is to 'normalize' the $s_\theta(x)$ output to make it in proper range. In this section, we test various methods for normalizing $s(x)$ to make it in bounded range. In details, we test 1) no_s where we omit $s(x)$ and only use $t(x)$. This is reported in Chao et al. (2024)

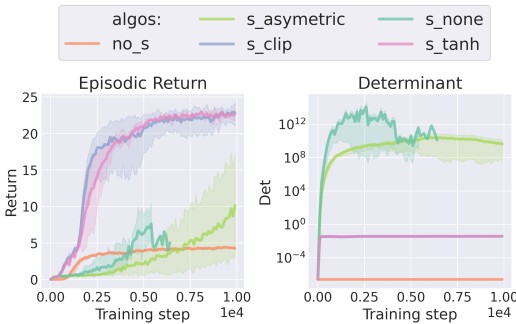

Figure 2: Training NFs in Unitree Gym's g1. The experiment is in 3 seeds with 95% confidence interval. s_none early stopped due to training instability, s_clip and s_tanh overlaps in determinant plot.

to have sufficient expressiveness in online RL setting, 2) s_clip where a $\text{clip}(s_\theta(x), -l, l)$ is applied with a hyperparameter $l$, 3) s_tanh where we use $l \times \tanh(s_\theta(x))$ and 4) s_asymetric which is reported in Andrade (2024) as an advanced 'normalizing' technique. We train all of these methods (with $l = 0.5$) with PPO in Unitree RL Gym's g1 environment, and the result is in Figure 2.

From the result, we find s_clip and s_tanh are 2 most effective 'normalizing' methods and both no_s and s_none obtains insufficient performance. While s_asymetric could obtain performance increase in the ending of training, the determinant of it is still in high scale. Finally, we choose s_tanh for its simplicity and robustness (in Section 5.1 we compare s_tanh and s_clip in experiments and provide an anlaysis over why tanh may be better than clip in Appendix A.1).

Finally, we could build a stabilized PPO-NF method which we name as NFPO (for brevity, the conditioned state $s$ is omitted):

$$\pi_{\text{new}} = \arg\max_\theta \mathbb{E}\Big[\min\big(r(\theta)\tilde{A}(s,a), \text{clip}(r(\theta), 1-\epsilon, 1+\epsilon)\tilde{A}(s,a)\big)\Big]$$

$$\log \pi(a) = \log q(f_\theta(a)) + \sum_j \log |\frac{df_{\theta_j}(a)}{da}| \qquad (6)$$

$$f_{\theta_j}(a)_{d_j} = a_{d_j}$$

$$f_{\theta_j}(a)_{\backslash d_j} = a_{\backslash d_j} \odot \exp(l \tanh(s_{\theta_j}(a_{d_j}))) + t_{\theta_j}(a_{d_j})$$

where $j$ represents the $j$th coupling layer. A pseudo-code of NFPO is presented in Appendix A.8.

## 5 EXPERIMENTS

In this section, we conduct experiments to answer several questions regarding NFPO's properties: 1) How do various factors influence NFPO's performance? 2) How is NFPO's performance compared to state-of-the-art (SOTA) PPO implementations? 3) Does NFPO learn multi-modal behaviors? 4) How is NFPO compared to other multi-modal policy learning methods? 5) Could NFPO's policy transfer to real-world robotic platforms? We conduct experiments in 2 widely used simulator: 1) Unitree RL Gym (URG) which is based on Nvidia IsaacGym (Makoviychuk et al., 2021) and is the official platform for training locomotion policies on Unitree's robots (g1, h1 and go2); 2) Mujoco Playground (MJP) (Zakka et al., 2025) which uses MJX as the underlying parallel simulation.

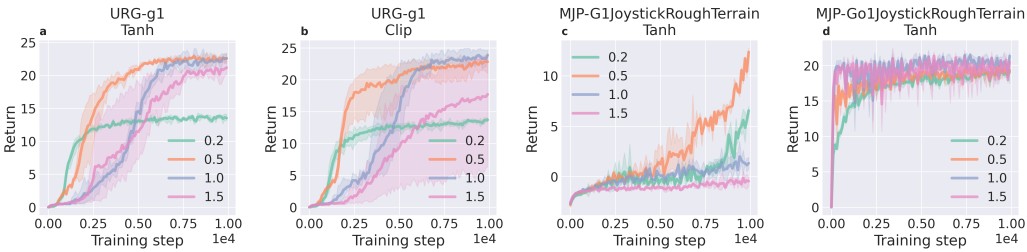

Figure 3: Studies in various factors. Experiments are in 3 seeds with 95% confidence interval.

### 5.1 Q1: HOW DO VARIOUS FACTORS INFLUENCE NFPO'S PERFORMANCE?

Our first experiment is to explore how various factors influence the NFPO's performance. As we have observed, the 'normalizing' technique is very important in stabilizing the training of NFs, we hence choose several related factors: 1) the hyperparameter $l$, 2) tanh vs clip normalization, 3) an entropy term that is used in PPO to prevent mode collapsing and 4) adding certain level of noise to actions during training then remove it in sampling phase, similar to Zhai et al. (2025).

We firstly investigate the influence of 'normalization' techniques between clip and tanh in Figure 3 (subfigures **a** and **b**): Generally speaking, both clip and tanh could obtain strong performance given proper $l$ values. However, tanh tends to be more robust towards $l$ than clip hence is chosen in our implementation.

Then we check the influence of $l$ on tanh by looking at subfigures **a**, **c** and **d** in Figure 3. In simpler environments like Go1JoystickRoughTerrain, various $l$ values could bring similar performance. But in challenging tasks like g1 and G1JoyStickRoughTerrain, proper value of $l$ plays a very important role in learning performance. From the results, we find $0.5$ is a good default value that fits various tasks.

Another important component in PPO is its entropy loss which could help exploration and prevent the methods from collapsing in local optimum. While NFPO is a multi-modal policy, we test if entropy term still helps its learning. By looking at results in Figure 4 (subfigures **a**, **c** and **e**), we could find adding entropy loss in our NFPO doesn't bring significant performance difference and could even slow down the learning speed in G1JoyStickRoughTerrain.

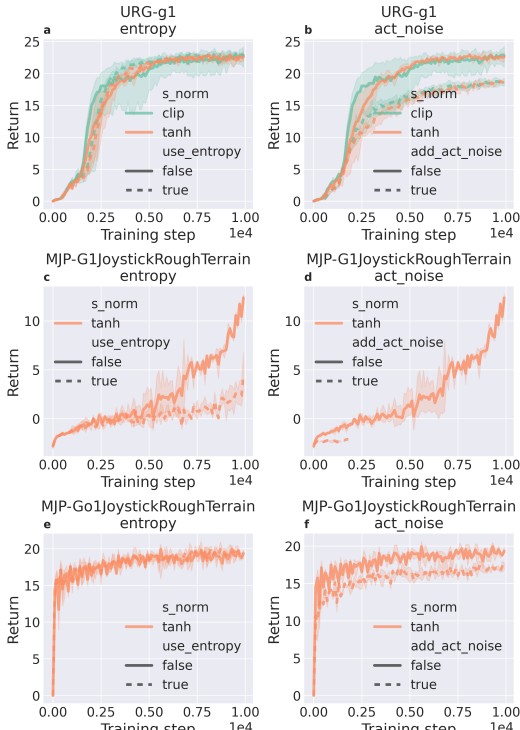

Figure 4: Studies in various factors. Experiments are in 3 seeds with 95% confidence interval.

Finally, as reported in Zhai et al. (2025), a critical technique to increase the image generation quality of NFs is to add Gaussian noise to training samples and remove this during generation, which is also observed in Ghugare & Eysenbach (2025). While our online RL settings have offered massive amount of continuous samples that could potentially reduce the need to 'dequantize' as in their dataset-based settings, we still test if adding this training noise and remove it during action sampling could help us further improve the performance. The results are in Figure 4 (subfigures **b**, **d** and **f**). Different from their settings, we find adding action noise would only decrease the RL performance in our tasks. Especially in some challenging environment like G1JoyStickRoughTerrain, it triggers training instability and early stops training at roughly 25% of total steps.

With above knowledge, we finally design our NFPO as a 4-layer RealNVP network with tanh transformations, remove entropy loss and use $l = 0.5$. **We use this configurations in all following experiments, replace Gaussian policy with it and find this obtains competitive performance without further tuning on other hyperparameters.**

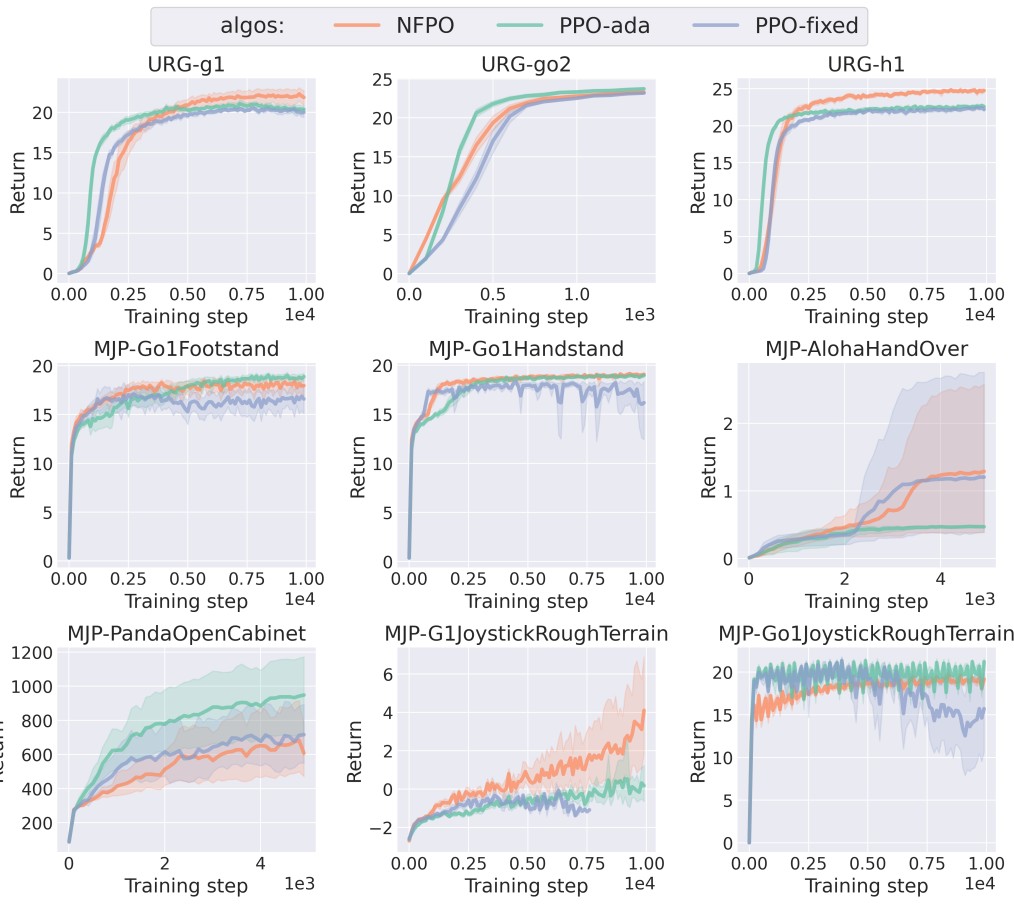

Figure 5: Learning curves of various methods on representative robotic tasks. The experiments are in 10 seeds. The errorbars are 95% confidence interval.

## 5.2 Q2: HOW IS NFPO'S PERFORMANCE COMPARED TO SOTA PPO IMPLEMENTATIONS?

In this section, we compare NFPO's performance against RSL-RL (Schwarke et al., 2025)'s PPO implementation, the latter is current SOTA method that widely used in real-world oriented policy learning in countless Robotics works. We compare NFPO to 2 variants of PPO: 1) PPO-adaptive where the learning rate is dynamically scheduled for faster convergence and 2) PPO-fixed where the learning rate scheduler is disabled and is the default setting in Mujoco playground. The experiments are run in 10 seeds and shown in Figure 5. For a fair comparison, we have aligned the actor network

size of `PPO` and `NFPO` and most other components (e.g., value network) and hyperparameters (e.g., learning rate) are shared. The result is in Figure 5. We run 4096 parallel environments for unitree rl gym and 2048 for mujoco Playground.

From the result, we observe `NFPO` could achieve competitive or stronger performance compared to `PPO` baselines. Specifically, in high-dimensional control tasks like g1-joystick, h1-joystick and G1JoyStickRoughTerrain, our `NFPO` could achieve stronger convergence performance, thanks to its multi-modal modeling ability. On Go1JoystickRoughTerrain, `NFPO`'s performance fluctuates in less extent, reflecting it's better to adapt to complex terrains. For simpler tasks like go2-joystick, `NFPO` still obtains similar performance against `PPO`. Finally, in manipulation tasks like AlohaHandOver and PandaOpenCabinet where exploration plays a more important part, the three tested methods exhibit complicated result with no one to be the best, indicating careful tuning is needed in these tasks. As for computation efficiency, on Unitree gym's g1 environment, we observed a roughly 19% increase of wall-clock time in `NFPO` with details in Appendix A.10. For tasks in mujoco playground, we provide a video of `NFPO`'s policies in simulation in supplemental materials.

### 5.3 Q3: Does NFPO learn multi-modal behaviors?

In this section, we test if `NFPO` could learn multi-modal behaviors given proper reward functions. Following works in FPO (McAllister et al., 2025), we use the gridworld example to demonstrate the difference in generated behaviors between `NFPO` and `PPO`. In gridworld environment, the agent is spawned in the grey cells and take continuous actions to move around. If it hits the green region, a positive reward would be granted otherwise the reward is 0.

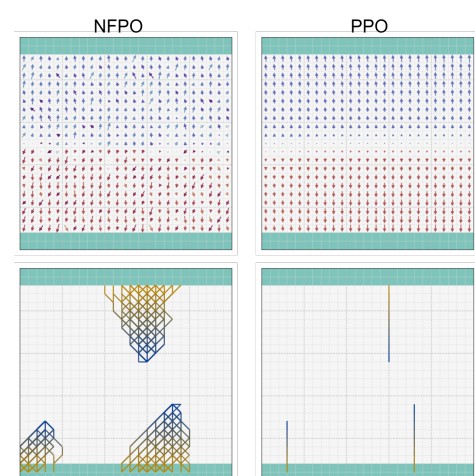

The result is shown in Figure 6. The upper 2 figures depict the actions each policy would take in certain position and the lower 2 figures depict the generated 100 trajectories from the same starting point of each method. In this sparse reward setting, we could find `NFPO` tends to generate diverse actions that lead to various trajectories towards green region while `PPO` only takes the shortest way and doesn't exhibit multi-modal behaviors.

Besides, we also train `NFPO` and `PPO` in Isaac-Reach-UR10-v0 environment in IsaacLab (Mittal et al., 2023). Specifically, we design a sparse-reward variant of UR10 reaching where the arm would gain a positive reward when reaching to the given target position. After training both `PPO` and `NFPO` to convergence, we fix the initial state and target, then run both `PPO` and `NFPO` for 100 rollouts, and plot their trajectories in Figure 7 where 'o' marks the initial position

Figure 6: Difference in generated trajectory of `NFPO` and `PPO` in gridworld.

(in xyz world frame) of end effector and 'x' marks the end position. We also attach the videos captured in IsaacLab's simulation environment to better visualize the behavior difference between `NFPO` and `PPO` in supplemental materials.

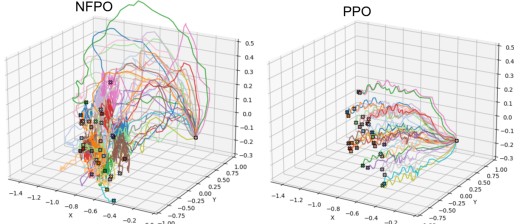

From the result, we could find `NFPO` generates diverse trajectories while `PPO` only runs in rather fixed trajectories.

### 5.4 Q4: How is NFPO compared to other multi-modal policy learning methods?

Figure 7: Difference in generated trajectories of `NFPO` and `PPO` in UR10 Reaching.

In this section, we try to compare `NFPO` against several related works that also employ multi-modal policy parameterizations in onlint RL setting. However, as also illustrated in Table 1, not all methods hold equal fitness and potentials to be used in robotic policy learning. Specifically, there are 2 lines of works that deserve a discussion and comparison:

1. On-policy diffusion-model-based methods like GenPO (Ding et al., 2025) and FPO (McAllister et al., 2025). The difference between NFPO to these methods mainly locate in the difference between diffusion-based method and normalizing flows: In NFs, we have closed-form, easily-calculated log probability that could be directly chained to PPO's training objective. While for diffusion models, various approximation methods need to be used and incur memory and computation overhead. As GenPO doesn't release their source code and is hungry in memory and computation resources, we compare against FPO in this genre of methods.

2. Off-policy normalizing-flows-based methods like Meow (Chao et al., 2024) and Ghugare & Eysenbach (2025). The difference between NFPO to these methods mainly locate in the training paradigm of PPO to off-policy methods: Off-policy methods usually have a larger replay buffer and better sample-efficiency but fall short of computation and memory efficiency as also studied in Seo et al. (2025). In this line of work, we compare against Meow as Ghugare & Eysenbach (2025) doesn't directly study conventional online RL settings.

As for other off-policy diffusion-based methods like MaxEntDP (Dong et al., 2025), QVPO (Ding et al., 2024) and QSM (Psenka et al., 2024), both their off-policy setting and approximation of log probability makes it non-trivial to integrate into normal training pipelines like IsaacGym and Mujoco Playground hence are omitted here.

In Figure 8, we present the training curves of Meow in Unitree RL gym's g1 and go2 environment. In Meow, it has a novel design where no explicit policy network exists. Rather, it uses value function $Q$ to parameterize the policy but this conflicts with the common asymmetric setting in robotic tasks where the value network and actor network have different observation dimensions and we have to feed actor observations (without privileged information) to its $Q$ and $V$ functions. Besides, we adjust its off-policy replay buffer's dimension to $(N_e, N_r, ...)$ where the first dimension is the number of environments (4096 or 2048), the second dimension is its original buffer length (1.5e4). Following FastTD3, we also increased the batch size per each update while all other settings are either from their original implementation or the same to PPO and NFPO (details are in Appendix A.4). Under this setting, it costs roughly 50GB GPU memory for training on 4096 unitree gym's g1 environment.

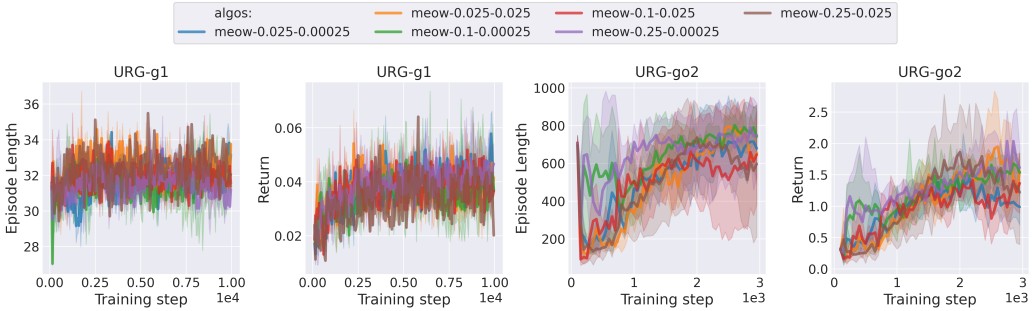

Figure 8: Learning curves of Meow and NFPO in 2 environments of Unitree RL Gym. Legend is meow-alpha-polyak where alpha and polyak are 2 tuned hyperparameters.

From the result, unfortunately, Meow doesn't exhibit meaningful performance in g1 and go2 environments. In g1, both the episode length and episodic return stay in low values while in go2, episode length achieves high values in some phase, indicating Meow's maximum-entropy learning framework could incentivize the exploration in some extent. However, the episodic return stays in low values and do not indicate a successful learning.

Table 2: NFPO and FPO's accomplished tasks in Mujoco playground.

| Task | FPO | NFPO |
|------|-----|------|
| AlohaHandOver | ✓ | ✗ |
| G1JoystickRoughTerrain | ✗ | ✓ |
| Go1Footstand | ✗ | ✓ |
| Go1Handstand | ✓ | ✓ |
| Go1JoystickRoughTerrain | ✗ | ✗ |
| PandaOpenCabinet | ✗ | ✓ |
| PandaPickCube | ✗ | ✓ |
| Total | 2 | 5 |

For FPO, we use their original codebase, and run the FPO's training with their default hyperparameters on 7 mujoco playground's tasks sim-

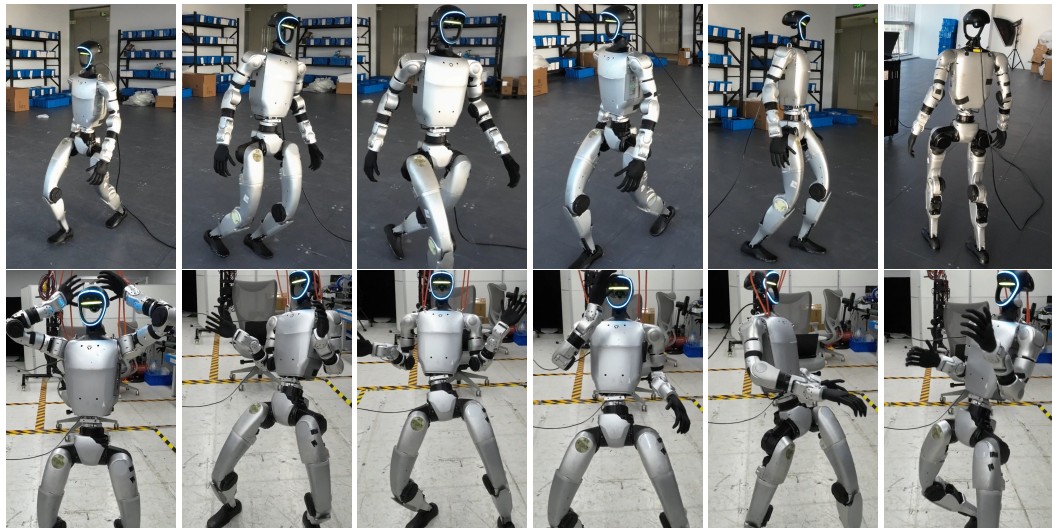

Figure 9: Snapshoots of real world deployment video.

ilar to what we used in Figure 5. Then we visualize the learned policies in official mujoco play-ground's codebase to see if they could successfully perform the tasks and show results in Table 2. For FPO, 2 out of 7 policies could show meaningful behaviors in simulation. And NFPO could perform 5 out of 7. Videos of both NFPO and FPO could be found in supplemental materials.

Integrating modern multi-modal policy into common robotic learning tasks is not a straightforward work, as the robotic tasks feature learning stability and memory and computation efficiency more than traditional RL tasks. And the purpose of this experiment is to showcase the learning stability and robustness of NFPO for which we use *a same set of configuration* across multiple simulation environments and tasks. It's also important to note this experiment doesn't mean FPO and Meow *won't work in robotic policy learning*. Actually, with proper hyperparameter tuning and training adjustment, it's likely for these methods to obtain improved performance, as in Seo et al. (2025).

### 5.5 Q5: COULD NFPO'S POLICY TRANSFER TO REAL-WORLD ROBOTIC PLATFORMS?

In this section, we transfer the policies trained by NFPO onto real-world robotic platform. In details, we choose Unitree RL Lab environment for training a joystick locomotion policy of and Beyond-Mimic (Liao et al., 2025) to train a dancing policy via motion-tracking. The robot is Unitree's g1 and we use the common hierarchical structure where a high-level RL policy (trained by NFPO) outputs action target then a low-level PD controller outputs joint torque accordingly. The high-level policy runs in 50Hz and the low level PD controller runs in 200Hz. The video of real world deployment is in supplemental materials and a series of the video clips is provided in Figure 9. Similar to Chao et al. (2024), we have found deterministic version of NFPO could generate stable and smooth real world motions than its stochastic counterpart (A discussion is provided in

## 6 CONCLUSION

In this work, aiming to unlock DRL-based multi-modal robotic policy learning, we explore the integration of Normalizing Flows to online Policy Optimization framework and have successfully built NFPO which is a stable method and could transfer to real-world robotic tasks.

In future works, we aim to further explore the benefits of NFPO to robotic policy learning, for eaxmple how to take advantages of its efficient calculation of log probability for interpretation and optimization of neural network based robotic policy.

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

# A APPENDIX

## A.1 ANALYSIS ON THE STABILITY OF TANH VS CLIP

In this section, we provide a theoretical analysis on why tanh may perform better than clip in mitigating the training instability.

Firstly, let's recall we are maximizing

$J(\theta) \approx \mathbb{E}[r(\theta)\tilde{A}(s,a)] \approx \mathbb{E}[\pi_\theta(a;s)] \approx \mathbb{E}\left[\exp\left(g\big(s_\theta(x)\big)\right)\right]$ where $g$ is one of *identity function* $(g(x) = x)$, *hard clip* $(g(x;l) = \mathrm{clip}(x,l))$ and *tanh* $(g(x;l) = l\tanh(x))$ ($l$ would be omitted for brevity in below).

If we analyze per-sample gradient of $J$ against $\theta$, we have:

$$G(x) = \nabla_\theta[\exp(g(s_\theta(x)))] = \exp(g(s))g'(s)\nabla_\theta s \quad \text{(S1)}$$

and the per-sample gradient scale is determined by multiplication of 3 factors:

1. $\exp(g(s))$: an exponential scale over $g$
2. $g'(s)$: how much the signal of outer exponential passes through
3. $\nabla_\theta(s)$: the neural network's sensitivity towards its parameters and are not influenced by the choice of $g$.

**Identity Function**. For $g(x) = x$, we have: $G(x) = \exp(s)\nabla(s)$ where $\exp(s)$ is unbounded and large $s$ could cause exploding gradients, same as what we observed in Figure 2.

**Hard Clip**. For $g(x;l) = \mathrm{clip}(x,l)$, we have: $G(x) = \exp(\mathrm{clip}(s))g'(s)\nabla(s)$. While it helps bound the exponential value, there are 3 minor issues related to it: 1) When $s$ is inside the range, the training signal is $\exp(s)$ which amplifies the gradient for larger $s$, creating a self-reinforcing push toward even larger values of $s$; 2) When s is outside clipped range, the gradient is 0, i.e. no training signal is received from this sample. Hence the optimizer loses ability to move those samples back; 3) Together, many samples are driven toward the boundary $|s| = l$, once they cross they become gradient-dead under hard clipping, which can stall or bias training.

**Tanh**. We have $G(x) = \exp(\tanh(s))(1 - \tanh^2 s)\nabla s$. Besides bounded first term $\exp(\tanh(s))$, another good property is the second term $(1 - \tanh^2 s)$ is also smoothly decreasing as $s$ increases. This helps optimizer to softly attenuate extreme values instead of zeroing them, so it avoids both exploding gradients and the "dead-zone" stalls caused by hard clipping.

## A.2 ADDITIONAL EXPERIMENTS ON MUJOCO

In this section, we compare `NFPO` to various classic online RL methods on Mujoco (Todorov et al., 2012) environments, we use `CleanRL` (**?**) as the testbed. For `SAC`, `TD3` and `PPO`, we use the original implementation and hyperparameters in `CleanRL`'s codebase. For `NFPO`, we just replace the policy parameterization from `PPO` and others remain the same. Note as `SAC`, `TD3` are off-policy methods, they generally obtain better sample efficiency compared to `PPO` and `NFPO`.

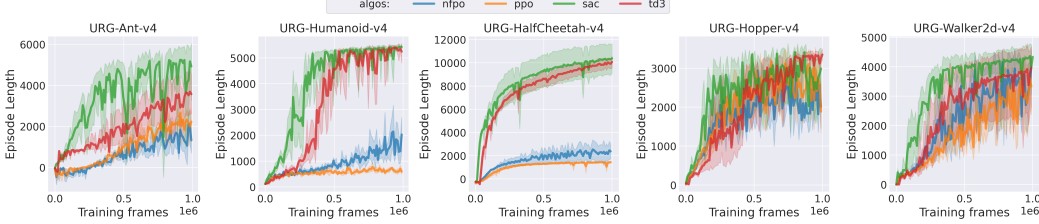

Figure S1: Performance of various online RL methods on Mujoco environments (5 seeds).

From the results, we could find `NFPO` could obtain similar performance compared to `PPO` on various simpler environments like Ant, HalfCheetah, Hopper and Walker2d. For complex environments like

Humanoid, it achieves better performance compared to `PPO`, similar to what we've found in the main experiments.

## A.3  ADDITIONAL ABLATION EXPERIMENTS OF `NFPO`

In this section, we further ablate on the *number of layers* and *hidden dimension size* of `NFPO`. The results are shown in Figure S2.

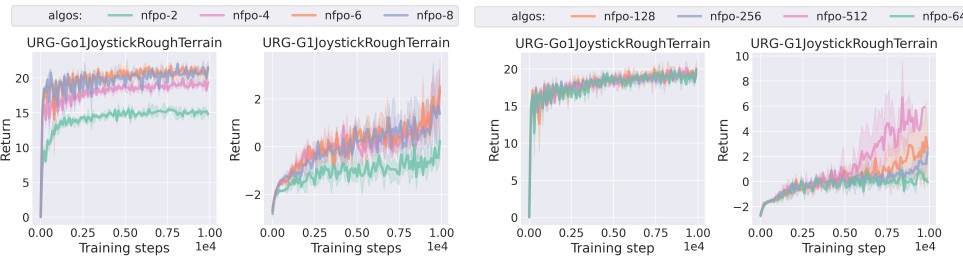

Figure S2: Performance of `NFPO` over various number of layers and hidden dimensions (5 seeds).

For number of layers, `nfpo-2` achieves significant lower performance. This is because for odd-even-based masking we used, minimum 3 layers are required for expressiveness. For other number of layers (e.g., 4, 6 and 8) , they generally obtain similar performance.

For hidden dimensions, they obtain quite similar results on simpler tasks like Go1JoystickRoughTerrain. For harder tasks like G1JoyStickRoughTerrain, `NFPO-512` obtains best performance while `NFPO-64` is the worst, and `NFPO-128` and `NFPO-256` obtain similar converged performance.

## A.4  ADDITIONAL TUNING OF `MEOW`

We additionally perform hyperparameters tuning for `Meow`. For the hyperparameters, we mainly tuned `polyak` and `entropy value` (`alpha`) similar to their original paper (Chao et al., 2024) and increased batch size to 10240 as recommended in Seo et al. (2025). Other hyperparameters are kept the same as in `Meow`'s original codebase. The results are in Figure S3 and the legend is meow-alpha-polyak (the tuned `alpha` values are {0.25, 0.1, 0.025} and `polyak` values are {2.5e-4, 2.5e-2}). To facilitate the reproducibility, we also provide the source code we used for training `Meow` in Unitree RL Gym.

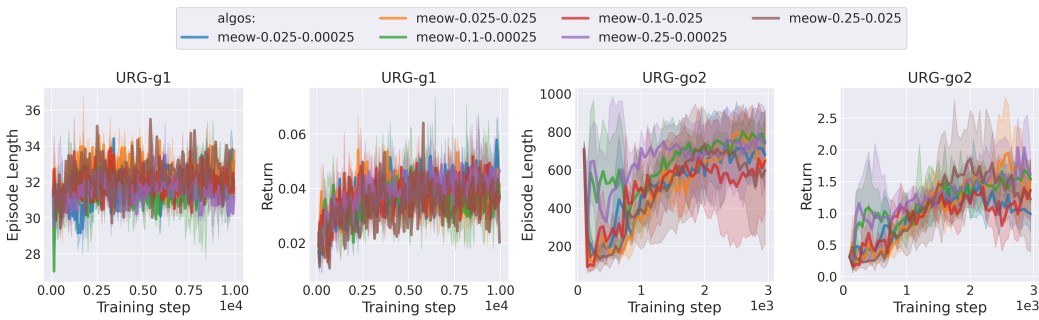

Figure S3: Performance of `Meow` over various tuned `polyak` and `alpha`. The experiments are in 3 seeds.

From the results, unfortunately, `Meow` generated no significant learning in these 2 environments. On URG-go2, its episode length is increased substantially, indicating it learns not to fall to ground to some extent but the episodic returns indicate that no meaningful locomotion or command following is learned. For URG-g1 which is more prone to fall, no significant learning in both episodic length and episodic return is observed.

## A.5 STABILIZED DEPLOYMENT

Deployment on real-world whole-body-controlled humanoid robot is complex and could cause damages to the surrounding environments or assets. Hence, in our experiment, we supply the *mode* of base Gaussian distribution to help stabilize the deployment. This is similar to the common practice in PPO where people use the *mean* ($\mu_\theta(s)$) as action in deployment (instead of sampling from $\mathcal{N}(\mu_\theta(s), \sigma_\theta(s))$ as in training).

This can also be seen as a similar case to what people do in temperature-based generation where high temperature favors more random generation while low temperature favors more optimal generation. For NFPO, temperature could be controlled in sampling $z \sim N(0, \tau I)$, for PPO, it's in $a \sim N(\mu_\theta(s), \tau\sigma_\theta(s))$. In our experiments, for both PPO and NFPO, if we sample actions with a larger temperature ($\tau$), both of them generate unstable behaviors. While using a low-temperature (a smaller $\tau$ that is not necessarily 0), both of them generate stable behaviors.

## A.6 THE MODE OF NORMALIZING FLOWS

For RealNVP:

$$f_{\theta_j}(a)_{d_j} = a_{d_j} \tag{S2}$$

$$f_{\theta_j}(a)_{\backslash d_j} = a_{\backslash d_j} \odot \exp(s_{\theta_j}(a_{d_j})) + t_{\theta_j}(a_{d_j}) \tag{S3}$$

$$\text{we have} \quad \log \pi(a) = \log q(f_\theta(a)) + \sum_j \log |\frac{df_{\theta_j}(a)}{da}| \tag{S4}$$

$$= \log q(f_\theta(a)) + \sum_j s_{\theta_j}(a_{d_j}) \tag{S5}$$

As it uses non-constant scale ($s_\theta(a)$), the mode in prior distribution ($\mathcal{N}(0, I)$) may not be the mode of data distributions. However, the $\log q(f_\theta(a))$ term still favors smaller latent and in practice we find sampling near the mode of prior distribution is sufficient to generate stable behavior as in Appendix A.5. This is also similar to what Kirichenko et al. (2020) finds that RealNVP tends to use $t$ to reduce the effect of an increased $s$ to get a smaller latent, as a way to meet the 2 objectives simultaneously.

In Chao et al. (2024), as they adopt NICE (Dinh et al., 2015) $\left(f_{\theta_j}(a)_{\backslash d_j} = a_{\backslash d_j} + t_{\theta_j}(a_{d_j})\right)$ which doesn't have varying scalar terms ($s_\theta(a) = 1$). Hence the mode in prior distribution is analytically the mode of data distribution.

## A.7 DETERMINATION OF VALUES IN TABLE 1

In this section, we explain how we choose the values for each algorithm presented in Table 1. We focus on explaining those that may cause confusion and omit the easy ones.

1. FastTD3 (Seo et al., 2025). In their paper, a critical modifications to the TD3 algorithm is to increase its batch size to a rather large value (roughly 32k). Combined with large off-policy replay buffer and massive environment numbers, the memory efficiency is sacrificed.

2. MaxEntDP (Dong et al., 2025). In their paper, to compute the log probability of diffusion policies (Eq. 21 in the original paper) and estimation of training target (Eq. 17 in the original paper), sampling-based approximation is employed. In high-dimensional space, the sampling number $N$ and $K$ is expected to be large and they use $N = 100$ and $K = 500$ as noted in their paper. In our setting, combined with large number of parallel environments (typical value 2048, 4096), this results in high consumption of memory.

3. GenPO (Ding et al., 2025). GenPO tries to compute the full Jacobian determinant without simplification used in Normalizing Flow and this brings much computation and memory burden. As noted in Conclusion and Limitation of their paper: '*Despite its excellent performance, GenPO faces the problem of relatively high computational and memory overhead to be resolved in the future.*'. And they used computing clusters with high memory and computation capacity as noted in their Appendix B.1.

The references: 1) FastTD3: (Seo et al., 2025); 2) Meow: (Chao et al., 2024) 3) MaxEntDP: (Dong et al., 2025); 4) GenPO: (Ding et al., 2025); 5) FPO: (McAllister et al., 2025)

## A.8 PSEUDOCODE OF NFPO

In this section, we provide a pseudo code of NFPO in Algorithm 1.

---

**Algorithm 1:** Pseudo code of NFPO

---

**Input:** parallel environment $e$, initial network parameter $\theta$, clip ratio $\epsilon$

1 **while** *training* **do**
2    $B \leftarrow []$
3    **for** $t = 1$ **to** $T$ **do**
4      $z \sim q(z)$
5      $a \leftarrow f_\theta^{-1}(z; o)$
6      $\log a = \log(f_\theta(a)) + \sum_j \log |\frac{df_{\theta_j}(a;o)}{da}|$
7      $o', r, d \leftarrow \texttt{step}(e, a)$
8      $B \leftarrow \texttt{append}(B, (o, a, r, d, \log a, o'))$
9    **end**
10    $\tilde{A} \leftarrow \texttt{GAE}(B)$ // estimate advantage via GAE
11    **for** $n = 1$ **to** $N$ **do**
12      $bs \leftarrow \texttt{partition}(B)$ // partition large buffer $B$ to smaller $bs$
13      **for** $m = 1$ **to** $M$ **do**
14        $b \leftarrow bs[m]$
15        $agent \leftarrow \texttt{update}(agent, b)$
16      **end**
17    **end**
18 **end**
19 **Function** $\texttt{update}(agent, b)$
     /* the update of value function is not changed and omitted */
20    $o, \pi_{\text{old}}(a) \leftarrow b$
21    $r(\theta) \leftarrow \frac{\pi_\theta(a;o)}{\pi_{\text{old}}(a)}$
22    $L_\theta \leftarrow \nabla_\theta \frac{1}{|b|} \sum_b \Big[ \min\big(r(\theta)\tilde{A}(s,a), \text{clip}(r(\theta), 1-\epsilon, 1+\epsilon)\tilde{A}(s,a)\big) \Big]$
23    $agent \leftarrow \texttt{SGD}(agent, L_\theta)$
24    **return** $agent$
25 **end**

---

## A.9 DETAILS OF HYPERPARAMETERS

We list the details of training hyperparameters in Table S1. For PPO and NFPO, most components and hyperparameters could be shared and major difference lies in the actor network parameterization which we have aligned in terms of parameter numbers.

## A.10 DETAILS OF HARDWARE AND COMPUTATION EFFICIENCY

Most of our experiments run on a 16 CPU x 1 Nvidia A100-40G GPU except for Meow which needs more than 40G GPU memory and run on A100-80G.

Depending on specific training settings, the wall clock time of each experiment varies and we provide the wall clock time of PPO and NFPO on Unitree RL Gym's g1 environment in Table S2:

Table S1: Hyperparameter settings

| Hyperparameter | | Value | Remarks |
|---|---|---|---|
| Shared | GAE $\lambda$ | 0.95 | |
| | discount $\gamma$ | 0.99 | |
| | value loss coefficient | 1.0 | |
| | grad clip | 1.0 | |
| | learning epoch | 5 | |
| | learning minibath | 4 | |
| | entropy loss coefficient | $10^{-3}$ | only for PPO |
| | step length | 24 | |
| | desired KL divergence | $10^{-2}$ | only for PPO adaptive |
| Unitree RL Gym | learning rate | $10^{-3}$ | |
| | value network hidden dims | $[32]$ | 1 layer network with hidden dim 32 |
| | PPO actor network hidden dimes | $[96, 96, 64]$ | |
| | NFPO actor network hidden dimes | $4 \times [64]$ | 4 layers, each layer with 64 hidden dim |
| | number of environments | 4096 | |
| Mujoco Playground | learning rate | $5e^{-4}$ | $3e^{-4}$ for manipulation tasks |
| | value network hidden dims | $[512, 256, 128]$ | |
| | PPO actor network hidden dimes | $[512, 256, 128]$ | |
| | NFPO actor network hidden dimes | $4 \times [256]$ | |
| | number of environments | 2048 | |

Table S2: Runtime of `PPO` and `NFPO` on Unitree Gym's g1. Calculated using 10 seeds.

| Algorithm | Runtime (s) | Runtime (h) | Ratio |
|---|---|---|---|
| PPO | $13500.94 \pm 628.29$ | $3.75 \pm 0.17$ | 1 |
| NFPO | $16036.06 \pm 795.74$ | $4.45 \pm 0.22$ | 1.19 |

## A.11 Detailed description of environments used in this paper

### A.11.1 URG-G1

A unitree g1 robot is trained to follow command for locomotion on ground. Note the upper body of it is fixed hence only 12 DoFs are used to control the lower body (2 legs, each with 6 DoFs).

actor observation space:

1. pelvis angular velocity: 3 dims

2. projected gravity: 3 dims

3. commands: 3 dims

4. joint velocity: 12 dims

5. (current_dof_pos - default_dof_pos): 12 dims

6. last actions: 12 dim

7. sin_phase: 1 dim

8. cos_phase: 1 dim

in total 47 dims

privileged state for critic:

1. pelvis linear velocity: 3 dims

2. pelvis angular velocity: 3 dims

3. projected gravity: 3 dims

4. commands: 3 dims

5. joint velocity: 12 dims

6. current_dof_pos - default_dof_pos: 12 dims

7. last actions: 12 dim

8. sin_phase: 1 dim

9. cos_phase: 1 dim

in total 50 dims

the actions are 12 dims positional control

reward settings:

1. Velocity Tracking: Rewards the robot for matching both commanded linear (forward/sideways) and angular (turning) velocities.

2. Stability & Orientation: Penalizes tilting away from an upright posture, deviations from a target base height, vertical velocity, and bod roll and pitch (wobbling).

3. Gait & Foot Movement: Rewards feet for being in the air and achieving a minimum swing height, rewards proper foot contact, and penalizes feet making contact while still moving horizontally (stumbling).

4. Energy & Effort Minimization: Penalizes high joint accelerations, high joint velocities, and large changes between consecutive actions fo smoother, more energy-efficient movements.

5. Behavior Regularization & Penalties: Penalizes joints approaching physical limits, collisions with the environment, and lateral hip displacement.

6. Survival: Provides a small, constant reward for each moment the robot has not fallen, encouraging longer episode duration.

### A.11.2 URG-H1

An unitree h1 robot is trained to following command for locomotion on ground. Note the upper body of it is fixed hence only 10 DoFs are used to control the lower body (2 legs, each with 5 DoFs).

actor observation space:

1. pelvis angular velocity: 3 dims

2. projected_gravity: 3 dims

3. commands: 3 dims

4. current joint position - default_joint_pos: 10 dims

5. current joint velocity: 10 dims

6. last step actions: 10 dims

7. sin_phase: 1 dims

8. cos_phase: 1 dim

in total 41 dims

privileged state for critic:

1. pelvis linear velocity: 3 dims

2. pelvis angular velocity: 3 dims

3. projected_gravity: 3 dims

4. commands: 3 dims

5. current joint position - default_joint_pos: 10 dims

6. current joint velocity: 10 dims

7. last step actions: 10 dims

8. sin_phase: 1 dims

9. cos_phase: 1 dim

in total 44 dims

the actions are 10 dims positional control

reward settgins:

1. Velocity Tracking: Rewards the robot for matching both commanded linear (forward/sideways) and angular (turning) velocities.

2. Stability & Orientation: Penalizes tilting away from an upright posture, deviations from a target base height, vertical velocity, and body roll and pitch (wobbling).

3. Gait & Foot Movement: Rewards feet for being in the air and achieving a minimum swing height, rewards proper foot contact, and penalizes feet making contact while still moving horizontally (stumbling).

4. Energy & Effort Minimization: Penalizes high motor torques, high joint accelerations, and large changes between consecutive actions for smoother, more energy-efficient movements.

5. Behavior Regularization & Penalties: Penalizes joints approaching physical limits, collisions with the environment, and lateral hip displacement.

6. Survival: Provides a small, constant reward for each moment the robot has not fallen, encouraging longer episode duration.

### A.11.3    URG-Go2

a unitree go2 legged robot is commanded to perform locomotions on ground, the robot has 4 legs, each with 3 DoFs.

observation space (for both actor and critic):

1. base linear velocity: 3 dims

2. base angular velocity: 3dims

3. projected_gravity: 3dims

4. commands: 3dims

5. current joint position - default joint position: 12 dims

6. current joint velocity: 12 dims

7. last step actions: 12 dims

in total 48 dims

the actions are 12 dims positional control.

reward settings:

1. Velocity Tracking: Rewards the robot for matching both commanded linear (forward/sideways) and angular (turning) velocities.

2. Stability & Orientation: Penalizes tilting away from an upright posture, deviations from a target base height, vertical velocity, body roll and pitch (wobbling), and any movement when commanded to be still.

3. Gait Foot Movement: Rewards feet for being in the air and penalizes feet making contact while still moving horizontally (stumbling).

4. Energy & Effort Minimization: Penalizes high motor torques, high joint accelerations, high joint velocities, and large changes between consecutive actions for smoother, more energy-efficient movements.

5. Behavior Regularization & Penalties: Penalizes joints approaching physical limits, collisions with the environment, and episode termination due to failure.

### A.11.4    MJP-AlohaHandOver

This environment has 2 aloha robotic arms and is tasked to transfer a box from one arm to another via cooperation. Each arm has 8 DoF where 6 are revolute joints (arm joints) and 2 are sliding joints (gripper).

Observation:

1. qpos: 23 dims joint position (2 x 8 + 3 + 4, the last 7 dims are cartesian position and quaternion of the box)

2. qvel: 22 dims joint velocity (2 x 8 + 3 + 3, the last 6 dims are linear and angular velocity of the box)

3. gripper's qpos - box_width: 4 dim

4. box_top coordinate: 3 dim

5. box_bottom coordinate: 3 dim

6. 3D coordinates of the left gripper in the world coordinate frame: 3 dim

7. The last 6 dimensions of the rotation matrix of the left gripper in the world coordinate frame: 6 dim

8. 3D coordinates of the right gripper in the world coordinate frame: 3 dim

9. The last 6 dimensions of the rotation matrix of the right gripper in the world coordinate frame: 6 dim

10. The last 6 dimensions of the rotation matrix of the box: 6 dim

11. The box's coordinates in the world frame minus target_pos (the predetermined transfer location): 3 dimensions

12. step: current step / episode length: 1 dim

Total: 83 dimensions

action: 14 dim positional control

reward settings:

1. gripper_box: Rewards the grippers for being close to the box.

2. box_handover: Rewards moving the box towards a predefined handover location.

3. handover_target: Rewards moving the box to the final target position with the right hand.

4. no_table_collision: A penalty is applied if the robot's hands collide with the table.

### A.11.5    MJP-PANDAOPENCABINET

A panda robotic arm to open a cabinet on it's handle

The panda arm has 7 revolute Dof and 2 sliding joints in gripper, and the handle of the cabinet is also a sliding joint.

1. qpos: joint position, 10 dim

2. qvel: joint velocity, 10 dims

3. The gripper's 3D world coordinates: 3 dims

4. The last 6 dimensions of the gripper's rotation matrix: 6 dims

5. The last 6 dimensions of the cabinet handle's rotation matrix: 6 dims

6. The difference between the cabinet handle's and the gripper's world coordinates (3D): 3dims

7. The difference between target_pos and the handle's 3D world coordinates: 3dims

8. The difference between the first 6 dimensions of the predefined target_mat rotation matrix and the first 6 dimensions of the handle's rotation matrix: 6dims

9. The difference between last-step action and the current joint positions: 8 dimensions

in total 55 dims

action: 8 dims positional control

reward:

1. gripper_box: Rewards the gripper for being close to the cabinet handle

2. box_target: Rewards moving the cabinet handle towards the target position. This reward is only active after the gripper has successfully reached the handle.

3. no_barrier_collision: The robot is rewarded for not colliding with a "barrier" object in the scene.

4. robot_target_qpos: This encourages the robot to stay close to its initial joint configuration, promoting more stable and predictable movements.

### A.11.6    MJP-G1JOYSTICKROUGHTERRAIN

A unitree G1 robot is commanded to move (joystick) on a rough terrain with domain randomization.

actor_obs:

1. noisy_linvel: 3D linear velocity with noise

2. noisy_gyro: gyroscope readings of the pelvis

3. noisy_gravity: IMU gravity readings of the pelvis, expressed in the base frame

4. command: 3D user command — x-velocity, y-velocity, yaw angular velocity (3 dimensions)

5. noisy_joint_angles - default_pos: 29 dimensions

6. noisy_joint_vel: 29 dimensions

7. phase: 4-dimensional phase representation (sine and cosine for the two legs)

in total 103 dims

other privileged states for critic:

1. Gyroscope: 3 dimensions

2. Accelerometer: 3 dimensions

3. Gravity in base frame: 3 dimensions

4. Linear velocity: 3 dimensions

5. Global angular velocity: 3 dimensions

6. joint_angles - default_pos: 29 dimensions

7. joint_vel: 29 dimensions

8. root_height: 1 dimension

9. actuator force: 29 dimensions

10. contact: 2 dimensions

11. feet_vel: 4 x 3 dimensions

12. feet air time: 2 dimensions

in total 216 dims

actions: 29 dims for whole-body control

reward:

1. Tracking: Rewards for matching the commanded linear and angular velocities.

2. Stability Penalties: Penalties for vertical velocity, rolling/pitching of the base, deviating from an upright orientation, and deviating from a target base height.

3. Energy Penalties: Penalties for high motor torques, large changes in actions (action rate), and overall energy consumption.

4. Gait & Foot Placement: A combination of rewards and penalties to encourage good gait quality, including rewards for foot air time and penalties for foot slippage, high contact forces, and self-collisions.

5. Pose Regularization: Penalties for deviating from the default joint pose, especially when commanded to stand still, and for getting too close to joint limits.

6. Termination Penalty: A large negative reward if the episode terminates due to falling or self-collision.

7. Alive Bonus: A small positive reward for each step the robot stays upright.

### A.11.7  MJP-GO1JOYSTICKROUGHTERRAIN

A unitree Go1 robot is commanded to move (joystick) on a rough terrain with domain randomization. It has 4 legs, each has 3 revolute joints.

actor obs:

1. noisy_linvel: 3D linear velocity with noise

2. noisy_gyro: 3D gyroscope readings

3. noisy_gravity: 3D gravity vector in the base frame

4. noisy_joint_angles - default_pose: 12 dimensions

5. noisy_joint_vel: 12 dimensions

6. last_act: 12 dimensions

7. command: 3D user command - x-velocity, y-velocity, yaw angular velocity

in total 48 dims

other privileged states for critic:

1. gyro: 3

2. accelerometer: 3

3. gravity: 3

4. linear velocity: 3

5. angular velocity: 3

6. joint_angles - default_pose: 12 dimensions

7. joint_vel: 12 dimensions

8. actuator_force: 12 dimensions

9. last_contact: 4 dimensions

10. feet_vel: 4 x 3 = 12 dimensions

11. feet_air_time: 4 dimensions

12. external disturbance force applied to the torso: 3 dimensions

13. whether the robot is being disturbed: 1 dimension

in total 123 dims

actions are 12 dimensional position control

reward:

1. Tracking: Rewards for matching the commanded linear and angular velocities.

2. Stability Penalties: Penalties for vertical motion of the base, excessive rolling and pitching, and deviating from a level orientation.

3. Energy Penalties: Penalties for high motor torques, rapid changes in action, and overall power consumption.

4. Gait & Foot Placement: A set of rewards and penalties to encourage a proper gait, including rewards for foot air-time and penalties for slippage and failing clear the ground during the swing phase.

5. Pose Regularization: A reward for staying close to the default pose and penalties for moving towards joint limits or deviating from the default pose while comanded to stand still.

6. Termination Penalty: A large negative reward if the episode ends from a fall.

### A.11.8 MJP-GO1HANDSTAND & MJP-GO1FOOTSTAND

the Go1 robot is commanded to raise its torso via only hand or foot.

state:

1. noisy_linvel: 3D linear velocity with noise

2. noisy_gyro: 3D gyroscope readings

3. noisy_gravity: 3D gravity vector in the base frame

4. noisy_joint_angles - default_pose: 12 dimensions

5. noisy_joint_vel: 12 dimensions

6. last_act: 12 dimensions

other privileged states for critic:

1. gyro: 3

2. accelerometer: 3

3. gravity: 3

4. linear velocity: 3

5. angular velocity: 3

6. joint angles - default_pose: 12 dimensions

7. joint_vel: 12 dimensions

8. actuator_force: 12 dimensions

9. torso_height: 1 dimension

reward settings:

1. 'height' (Reward): Encourages the robot's torso to reach a target height of 0.55 meters.

2. 'orientation' (Reward): Rewards the robot for being upside down, with its belly pointing directly at the floor.

3. 'contact' (Penalty): Applies a penalty if the front (or rear) feet make contact with the floor. This forces the robot to balance only on its rear (or front) feet.

4. 'pose' (Penalty): Penalizes the rear (or front) leg joints for deviating from a default pose, encouraging them to remain stable.

5. 'stay_still' (Penalty): Penalizes movement in the horizontal plane (both linear and turning), encouraging a stationary stand.

6. Common Penalties: Both environments also include penalties for joints getting too close to their limits (dof_pos_limits), high motor torques (torques), and rapid changes in motor commands (action_rate).

