# OpenReview forum: "NFPO: Stabilized Policy Optimization of Normalizing Flow for Robotic Policy Learning"
_ICLR.cc/2026/Conference — Submitted to ICLR 2026_

### Official Review · Reviewer_AJ7p · 2025-10-28

**Soundness:** 2
**Presentation:** 2
**Contribution:** 2
**Rating:** 4
**Confidence:** 5

**Summary:**

The paper proposes NFPO, an on-policy RL method that replaces the standard Gaussian policy with a normalizing flow (specifically, RealNVP) inside PPO. The authors experimentally analyze why naïvely plugging a flow into PPO is unstable, and proposed techniques to stabilize training by bounding the scale network with a tanh transform and a limit parameter l. Experiments span URG and MuJoCo Playground, and compare NFPO against PPO baselines. NFPO often matches or beats PPO on several locomotion/manipulation tasks; qualitative studies (gridworld and UR10 reaching) suggest more multi-modal behavior; and they show sim-to-real transfers to Unitree robots. Wall-clock cost increases by ~19% vs. PPO on g1.

**Strengths:**

- The proposed technique is simple and clearly presented, with writing that is easy to follow.
- The paper includes detailed studies on different design choices (e.g., different methods to stablize scaling function, different hyper-parameter choices).
- NFPO offers slight performance gains compared to PPO, especially for certain control tasks such as g1-joystick, h1-joystick, and G1JoyStickRoughTerrain.

**Weaknesses:**

- While the paper may be among the first to pair normalizing flows with on-policy RL, similar ideas have been explored extensively in off-policy settings, e.g., [1-3]. Without a clearer technical distinction or theoretical contribution beyond the training stabilizations, the novelty of this work seems modest.

- Reported gains are small and the comparisons are not run on widely accepted benchmarks with commonly adopted baselines (e.g., mixture Gaussian PPO variants, off-policy methods). As a result, it’s hard to assess the practical significance of the method.

- The sim-to-real experiments omit key operational specifics, e.g., task definitions, control frequency/latency, domain-randomization settings, and failure rates, making reproducibility and reliability difficult to judge.

---

**Minor Errors:**
- “the $\exp(s_\theta (x_d))$ used in RealNVP apply …” → “the $\exp(s_\theta (x_d))$ used in RealNVP applies …” in Line 171.
- “overfiting” → "overfitting"; appears in Lines 162-174.
- “logprobability” → "log-probability" (hyphen) in Line 177.

**Questions:**

- Please provide the state/action space definitions and concise descriptions of each evaluated environment. Also explain why NFPO strongly outperforms PPO on G1JoyStickRoughTerrain yet underperforms on MJP-PandaOpenCabinet—what properties of these tasks favor or hinder flow policies?

- Please include learning curves on common MuJoCo benchmarks (e.g., Hopper, HalfCheetah, Walker2d, Ant, Humanoid) and compare against widely used baselines (e.g., SAC, TD3) to contextualize performance.

- The experiment presented in Fig. 8 appears to modify baseline defaults without retuning. Because online RL is sensitive to hyperparameters, fair comparisons should use the best settings found within a shared search space. How were the baseline hyperparameters chosen, and why is that selection representative?

- Section 5.5 claims NFPO supports deterministic action sampling, but for non-volume-preserving flows the mode (argmax density) is generally nontrivial [3]. How is deterministic sampling implemented for RealNVP in practice?

- Fig. 5 suggests NFPO and PPO achieve similar returns, while Table S2 reports ~19% sampling overhead for flows. Under what conditions (task characteristics, exploration regime, multimodality) are normalizing-flow policies clearly preferable than Gaussian policies?

- Beyond the reported settings, how sensitive is performance to flow depth (e.g., 2/6/8 layers) and hidden sizes? Please include a brief ablation.

- Did you evaluate other efficient multimodal policy families (e.g., consistency models [4,5] or related diffusion variants)? How do they compare to NFPO in stability, sample efficiency, and runtime?

---

**References:**

[1] Haarnoja et al. Latent space policies for hierarchical reinforcement learning. ICML 2018. \
[2] Mazoure et al. Leveraging Exploration in Off-policy Algorithms via Normalizing Flows. CoRL 2019. \
[3] Chao et al. Maximum Entropy Reinforcement Learning via Energy-Based Normalizing Flow. NeurIPS 2023. \
[4] Song et al. Consistency Models. ICML 2023.\
[5] Ding et al. Consistency Models as a Rich and Efficient Policy Class for Reinforcement Learning. ICLR 2024.

---

> ### Author Response · Authors · 2025-11-21
>
> W1: *While the paper may be among the first to pair normalizing flows with on-policy RL, similar ideas have been explored extensively in off-policy settings, e.g...*
>
> R1:
> As noted in General Response 2 and discussed in FastTD3 papers, integrating off-policy rl methods into on-policy learning scenarios is a non-trivial work. Hence we don't think existing off-policy works overshadow the contribution of our work.
>
> $~$
>
> W2: *Reported gains are small and the comparisons are not run on widely accepted benchmarks...*
>
> R2:
> 1. as noted in General Response 1, in all the conducted experiments, we use the same hyperparameters without any tuning. For ppo and nfpo, the value network is also shared the same. In this case, it's reasonable for the results tend to be similar. However, under 10 seeds evaluation, our results show advantageous performance for NFPO on complex environments and tasks like URG-H1, URG-g1 and MJP-G1JoystickRoughTerrain.
> 2. we have provided additional experiments showing NFPO also achieves advantageous performance on Mujoco baseline in Appendix A.2
>
> $~$
>
> W3: *The sim-to-real experiments omit key operational specifics, e.g., task definitions, control frequency/latency, domain-randomization settings, and failure rates, making reproducibility and reliability difficult to judge.*
>
> R3:
> As discussed in General Response 1, we use all the default parameters and settings(reward settings, control frequence or domain-randomization) in all experiments conducted (including sim2real training and deployment), and have provided a detailed description for URG and MJP environments in appendix A.9.
>
> for sim-to-real experiments, the unitree RL Lab could be seen as a improved version of Unitree RL Gym  (e.g., the G1 is now whole-body-controlled and several reward functions, domain randomization techniques, observation shape may be improved) but the general pipeline is similar.
>
> for BeyondMimic, please refer to their own paper as it requires excessive space to summarize their settings and design here
>
> $~$
>
>
> Minor Errors:
> “the  used in RealNVP apply …” → “the  used in RealNVP applies …” in Line 171.
>
> “overfiting” → "overfitting"; appears in Lines 162-174.
>
> “logprobability” → "log-probability" (hyphen) in Line 177.
>
> $~$
>
> We really appreciate the reviewer's suggestions and have revised our manuscript accordingly.

---

> ### Author Response · Authors · 2025-11-21
>
> Q1: *Please provide the state/action space definitions and concise descriptions of each evaluated environment. Also explain why NFPO strongly outperforms PPO on G1JoyStickRoughTerrain yet underperforms on MJP-PandaOpenCabinet—what properties of these tasks favor or hinder flow policies?*
>
> R1:
> 1. please see Appendix A.9 for details of each environment
>
> 2. PandaOpenCabinet and AlohaHandOver are 2 manipulations where the  control difficulty may be less important ( as their root are bound to the table) than exploration (this is also why in manipulation policy learning, supervised learning is widely employed to avoid the exploration difficulty or reward engineering). For example, in AlohaHandOver, 2 hands are commanded to  cooperatively move a box to a far destination.   In this case, if the policy could discover the critical relaying behavior, then the trajectory would be optimal. Otherwise it may fall into local minimum(as shown in the video of NFPO). However, for G1 or Go1 legged robots' controlling, it's more like a long-term continual learning process where the robot needs to learn multiple behaviors sequentially during the learning process to stand up, stand still, follow varying commands and adapt to rough terrain and domain randomization. This is where multi-modality and modeling capacity comes into effect and our NFPO performs better than PPO, as expected.
>
> $~$
>
>
> Q2: *Please include learning curves on common MuJoCo benchmarks (e.g., Hopper, HalfCheetah, Walker2d, Ant, Humanoid) and compare against widely used baselines (e.g., SAC, TD3) to contextualize performance.*
>
> R2:
> We have provided an additional experiments showing NFPO's performance in Mujoco environments in appendix A.2.
>
> $~$
>
> Q3: *The experiment presented in Fig. 8 appears to modify baseline defaults without retuning. Because online RL is sensitive to hyperparameters, fair comparisons should use the best settings found within a shared search space. How were the baseline hyperparameters chosen, and why is that selection representative?*
>
> R3:
> We have actually spent much effort checking our meow implementation and have conducted an additional tuning on it (results are in Appendix A.4). We have also uploaded our codebase for training meow in URG with detailed and easy steps to show the modifications made by us.
>
>
> $~$
>
> Q4: *Section 5.5 claims NFPO supports deterministic action sampling, but for non-volume-preserving flows the mode (argmax density) is generally nontrivial [3]. How is deterministic sampling implemented for RealNVP in practice?*
>
> R4:
> yes, for non-volume-preserving flows the mode is generally non-trivial as discussed in [3]. But from the definition of RealNVP, the training objective has 2 parts:
>
> $ \log \pi(a) = \log q(f_\theta(a)) + \sum\nolimits_j \log \vert  \frac{df_{\theta_j}(a)}{da} \vert  = C - \frac{1}{2} \vert \vert f_{\theta}(a) \vert \vert^2 + \sum \log \Big| \frac{df_{\theta_j}(a)}{da_j} \Big| $
>
> While the jacobian term may  assign various log probability to different points, the gaussian term still favors small-norm $z$ (as in [1], a common learning pattern of RealNVP is to use $t_\theta(x)$ to reduce the norm of transformed samples hence  $s_\theta(x)$ could be increased to large value such that log probability is increased for both terms).
>
> As humanoid robot is a very complex robotic platform that may cause damages, in practice we use zero z  to help further stabilize the performance of NFPO in deployment. This is the same as in PPO where it's a common practice to pass the mean (instead of   actions sampled from normal)  into underlying PD controller.
>
> This can also be seen as a similar case to what people do in  temperature-based generation where high temperature favors more random generation while low temperature favors more optimal generation. For nfpo, temperature could be controlled  in sampling $z \sim N(0, \tau I)$, for ppo, it's in $a \sim N(\mu_\theta(s), \tau \sigma_\theta(s))$. In our experiments, for both ppo and nfpo, if we sample actions with a larger temperature ($\tau$), both of them generate unstable behaviors. While using a low-temperature (a smaller $\tau$ that is not necessarily 0), both of them  generate stable behaviors.
>
> >[1] Kirichenko, Polina, Pavel Izmailov, and Andrew G. Wilson. "Why normalizing flows fail to detect out-of-distribution data." Advances in neural information processing systems 33 (2020): 20578-20589.

---

> ### Author Response · Authors · 2025-11-21
>
> Q5: *Fig. 5 suggests NFPO and PPO achieve similar returns, while Table S2 reports ~19% sampling overhead for flows. Under what conditions (task characteristics, exploration regime, multimodality) are normalizing-flow policies clearly preferable than Gaussian policies?*
>
> R5:
> the critical difference between NFPO and PPO lays in the modeling capacity (multi-modal ability). As shown in our Sect. 5.2  for simpler tasks, NFPO obtains similar performance to SOTA PPO and better performance on harder tasks. Also in Sect 5.3, NFPO could generate more diverse trajectories to fullfil the tasks.
>
> Also, please note PPO-gaussian is a quite fast method. And we argue for multi-modal policies like diffusions, flow models, our NFPO is already among the top tiers in terms of  speed with just 19% increase in computation overhead.
>
> $~$
>
> Q6: *Beyond the reported settings, how sensitive is performance to flow depth (e.g., 2/6/8 layers) and hidden sizes? Please include a brief ablation.*
>
> R6:
> please see newly-added appendix in A.3
>
> $~$
>
> Q7: *Did you evaluate other efficient multimodal policy families (e.g., consistency models [4,5] or related diffusion variants)? How do they compare to NFPO in stability, sample efficiency, and runtime?*
>
> R7:
>  we have explored other existing rl-based multi-modal policy learning methods in our preliminary experiments: 1. iterative bc methods like [1] 2. diffusion-based methods like [2].
>
> >[1] Yang, Long, et al. "Policy Representation via Diffusion Probability Model for Reinforcement Learning." arXiv e-prints (2023): arXiv-2305.
> >
> >[2]  Dong, Xiaoyi, Jian Cheng, and Xi Sheryl Zhang. "Maximum Entropy Reinforcement Learning with Diffusion Policy." Forty-second International Conference on Machine Learning.
>
> Unfortunately, most of them fall short of either generating signficant performance in real-world deployment oriented learning environments like unitree rl gym, or require extensive memory or computation overhead (as also compared in our Table 1). The reasons are multi-fold:
> 1. many of these methods are off-policy based, and it has been shown to be non-trivial to transfer off-policy methods to on-policy sceranio as in FastTD3 [4], not to mention more complex multi-modal diffusion or flow methods.
> 2.  many of these methods rely on approximate methods to compute log-probability, while it may work in off-policy settings, in on-policy PPO based settings, this approximation usually brings training instability
>
> Among them, we have spent considerable time checking and tuning Meow as our baseline. However, no significant performance is obtained even after several rounds of tuning (not limited to the alpha and polyak as tuned in Appendix A.4)
>
> To summarize, this is why we think the findings of NFPO is beneficial for the community as it requires very minimum tuning effort to obtain decent performance across robotic policy learning framework ( see our General Response 1, for all experiments in this work, we just replace the policy of gaussian to NFPO then the training would succeed to very decent performance). This is in contrast to existing multi-modal RL learning methods which may require heavy tuning and sensitive towards some critical hyperparamters.

---

> > ### Comment · Reviewer_AJ7p · 2025-11-24
> >
> > Thank you for the detailed response. I appreciate the thorough experiments, and most of my concerns have been addressed. I have a couple of follow-up questions regarding Q3 and Q4:
> > - As stated in Line 398, the value network and actor network have different observation dimensions, which conflicts with the setup used in [3]. In this case, I think that a joint modeling approach may underperform and may not be an ideal baseline for this experiment. Could the authors instead compare NFPO with other similar methods (e.g., SAC)?
> > - Regarding policy inference, I believe a more detailed discussion of the theoretical explanation is necessary. I would suggest adding a dedicated section analyzing under what conditions one can guarantee that the mode can be derived efficiently and accurately. This would strengthen the theoretical grounding and clarify the applicability of the proposed approach.

---

> > > ### Author Response · Authors · 2025-11-27
> > >
> > > We really appreciate  your  reply and are happy that we have clarified some of your concerns.
> > >
> > > $~$
> > >
> > > Q8: *As stated in Line 398, the value network and actor network have different observation dimensions, which conflicts with the setup used in [3]. In this case, I think that a joint modeling approach may underperform and may not be an ideal baseline for this experiment. Could the authors instead compare NFPO with other similar methods (e.g., SAC)?*
> > >
> > >
> > > R8: As discussed in Section 5.4 Q4: *how is NFPO compared to other multi-modal policy learning methods*?
> > >
> > > We mainly choose 2 streams of multi-modal policy methods for comparison: 1. other normalizing flow-based method that also has accurate log-probability. In this stream, there are mainly 3 works [1-3]. Unfortunately, [1] doesn’t release its source code, [3] doesn’t directly study online rl setting, so we mainly choose Meow [2] for comparison.
> > >
> > > For the second stream, where general flow/diffusion methods are employed  without accurate log-probability calculation (Hence they either don’t use policy optimization or use approximation), we have actually tried many methods, e.g. [4-6]. Unfortunately, as also noted in our reply to Q7, [4,5] fail to generate meaningful learning results even after tuning.
> > >
> > > And FPO is the most performant and recent method so we give a detailed comparison to it in Sect 5.4.  (Actually, by using their *own implementation and hyperparameter*, we find it also involves nan values (also discussed in https://github.com/akanazawa/fpo/issues/4) when  training on some envs, but  we chose to ignore the nans and manually take their best checkpoints for  comparison. )
> > >
> > > This is why we summarized in Sect. 5.4: “*Integrating modern multi-modal policy into common robotic learning tasks is not a straightforward work, as the high-dimensional robotic tasks feature learning stability and memory and computation efficiency more than traditional RL tasks.*”. Based on our experience, there is a clear performance degradation when integrating existing multi-modal RL methods into common robotic learning frameworks like legged gym or mujoco playground. And this is why we think the stability and easy-to-tune of NFPO is valuable to the community.
> > >
> > > For other conventional methods like TD3 or SAC, we have provided a comparison in common mujoco in A.2. For their performance in robotic learning environments, we  recommend FastTD3[7] for a thorough discussion. Last but not least, the biggest difference between FastTD3/FastSAC and NFPO lies in the multi-modal learning ability of policy parameterization.
> > >
> > > > [1] Mazoure et al. Leveraging Exploration in Off-policy Algorithms via Normalizing Flows. CoRL 2019.
> > > /.
> > > > [2] Chao, Chen-Hao, et al. "Maximum entropy reinforcement learning via energy-based normalizing flow." Advances in Neural Information Processing Systems 37 (2024): 56136-56165.
> > > >
> > > > [3] Ghugare, Raj, and Benjamin Eysenbach. "Normalizing Flows are Capable Models for RL." arXiv preprint arXiv:2505.23527 (2025).
> > > >
> > > > [4] Yang, Long, et al. "Policy Representation via Diffusion Probability Model for Reinforcement Learning." arXiv e-prints (2023): arXiv-2305.
> > > >
> > > > [5] Dong, Xiaoyi, Jian Cheng, and Xi Sheryl Zhang. "Maximum Entropy Reinforcement Learning with Diffusion Policy." Forty-second International Conference on Machine Learning.
> > > >
> > > > [6] McAllister, David, et al. "Flow matching policy gradients." arXiv preprint arXiv:2507.21053 (2025).
> > > >
> > > > [7] Seo, Younggyo, et al. "FastTD3: Simple, Fast, and Capable Reinforcement Learning for Humanoid Control." arXiv preprint arXiv:2505.22642 (2025).
> > >
> > >
> > >
> > > $~$
> > >
> > > Q9: *Regarding policy inference, I believe a more detailed discussion of the theoretical explanation is necessary. I would suggest adding a dedicated section analyzing under what conditions one can guarantee that the mode can be derived efficiently and accurately. This would strengthen the theoretical grounding and clarify the applicability of the proposed approach.*
> > >
> > > R9:
> > > This is definitely a good suggestion, we have added 2 new appendix: A.5 (Stabilized Deployment) and A.6 (The mode of Normalizing Flows) to further explain this point. Please refer to the newly-revised pdf for details.
> > >
> > > For any further questions, please feel free to drop them and we’re happy to discuss

---

### Official Review · Reviewer_47UP · 2025-10-31

**Soundness:** 4
**Presentation:** 3
**Contribution:** 2
**Rating:** 6
**Confidence:** 3

**Summary:**

This paper aims to integrate the multi-modal modeling capabilities of Normalizing Flows (NF) into robotic PPO policies. The authors first identify the training instability of combining NF directly with PPO, attributing it to the exploding Jacobian determinant caused by the $exp(s)$ term in RealNVP.  To solve this, the authors propose NFPO, which uses an ${tanh}$ activation function to constrain the output of $s$, thereby stabilizing training. Experiments show that NFPO performs robustly on several simulation tasks, matching or exceeding  PPO 3, and was successfully transferred to a real-world robot.

**Strengths:**

Problem Diagnosis and Solution: The paper clearly diagnoses the root cause of instability when combining NF with PPO (exploding determinant) and proposes a simple, effective solution ($tanh$ activation) .

Solid Experimental Validation: The paper provides comprehensive benchmarks on 9 tasks across multiple simulators (IsaacGym, Mujoco Playground) 7and includes thorough ablation studies.

Real-World Deployment: The policy was successfully transferred from simulation to a real Unitree G1 robot, strongly demonstrating the algorithm's robustness and practical value.

**Weaknesses:**

Limited Innovation: The work is primarily an application and engineering-level adaptation of RealNVP for policy optimization, rather than a fundamental algorithmic innovation. The core stabilization technique ($s\_tanh$) is a known trick.

Ambiguous Multi-modal Advantage: Although multi-modality was shown in specific tasks (Sec 5.3), its direct link to the performance gains in the main benchmarks (Sec 5.2) is not clear.

**Questions:**

Regarding Real-World Deployment: You mentioned using a "deterministic version" in Sec 5.5.a) How was this "deterministic version" obtained (e.g., $z=0$)? Does this imply that the stochastic or multi-modal policy is unstable in the real world?

Regarding FPO/Meow Comparison: In Sec 5.4, NFPO outperformed FPO and Meow in robustness. Were FPO/Meow given the same level of hyperparameter tuning as NFPO, or were their default parameters used?

---

> ### Author Response · Authors · 2025-11-21
>
> W1: *Limited Innovation: The work is primarily an application and engineering-level adaptation...*
>
> R1: please see General Response 2
>
> $~$
>
> W2: *Ambiguous Multi-modal Advantage: Although multi-modality was shown in specific tasks (Sec 5.3), its direct link to the performance gains in the main benchmarks (Sec 5.2) is not clear.*
>
> R2:
> As also discussed in General Response 1, in the main experiments (Sect. 5.2), NFPO obtains significant performance gains in complex environments like H1, g1  and Go1JoystickRoughTerrain, in 10 seeds, compared to heavily-tuned, deployment-ready PPO implementation, under the same hyperparamters.  This illustrates its modeling capacity helps policy learning in these complex scenarios.
>
> For a detailed, rigiours analysis into the interplay between multi-modal policy and complex humanoid robotic tasks, we hope to leave it as future work as to our knowledge, currently no work or approach is developed for this yet.
>
> $~$
>
> Q1: *Regarding Real-World Deployment: You mentioned using a "deterministic version" in Sec 5.5.a) How was this "deterministic version" obtained (e.g., )? Does this imply that the stochastic or multi-modal policy is unstable in the real world?*
>
> R3:
> This is a very good question. As real-world humanoid robot is very complex, prone to fall and can cause damages, we use zero z  to help further stabilize the performance of NFPO in deployment. This is the same as in PPO, it's common practice to pass the mean (instead of  action sampled from normal)  into underlying PD controller.
>
> However, we don't think this means stochastic or multi-modal policy is unstable in the reward world.  Actually, this can be seen as a similar case to what people do in  temperature-based generative sampling where high temperature favors more random generation while low temperature favors more optimal generation. For nfpo, temperature could be controlled  in sampling $z \sim N(0, \tau I)$, for ppo, it's in $a \sim N(\mu_\theta(s), \tau \sigma_\theta(s))$. In our experiments, for both ppo and nfpo, if we sample actions with a larger temperature ($\tau$), both of them generate unstable behaviors. While using a low-temperature (a smaller $\tau$ that is not necessarily 0), both of them  generate stable behaviors.
>
> For simpler embodiment like robotic arms, multi-modal policy is already widely used and deployed in real world.
>
> $~$
>
> Q2: *Regarding FPO/Meow Comparison: In Sec 5.4, NFPO outperformed FPO and Meow in robustness. Were FPO/Meow given the same level of hyperparameter tuning as NFPO, or were their default parameters used?*
>
> R4: We take no hyperparameter tuning for all of these, please see General Response 1
>
> For FPO , we use their default parameter and original codebase, which is already tuned by the original authors
>
> For meow, we have also tuned it in newly-conducted experiments, please see Appendix A.4. We have also uploaded our code of meow (with detailed and easy steps to show modiciations we make) for the sake of reproducibility

---

### Official Review · Reviewer_MJyh · 2025-11-01

**Soundness:** 2
**Presentation:** 3
**Contribution:** 2
**Rating:** 2
**Confidence:** 3

**Summary:**

This paper proposes NFPO algorithm, which parameterizes the policy-network with a Normalizing Flow to capture the multi-modal action distributions. The authors diagnose the instability in RealNVP-based flows inside PPO, and provide a simple but effective solution: normalizing the $s_θ(x)$ output to make it in a proper range. Extensive experiments on multiple robotics simulators and with multiple seeds show a stronger and more stable performance on several locomotion tasks, with mixed results on some manipulation tasks and sim-to-real deployments on Unitree hardware.

**Strengths:**

- The authors propose NFPO, a new framework that integrates Normalizing Flows (NF) into PPO for robotic multi-modal policy learning, and further analyze the causes of its training instability and introduce effective stabilization techniques. The authors provide a clear problem formulation for the multi-modal action distribution for on-policy control, and a simple and reproducible solution by swapping policy head and bounding the scale output of the flow.

- Comprehensive experiments are conducted on several widely used simulation environments. With the same configuration settings, NFPO achieves competitive performance compared with state-of-the-art Gaussian-based PPO implementations. Real-world validation also demonstrates that policies trained with NFPO can be successfully transferred to physical robots. These extensive experiments on multiple robotics simulators and deployments show the effectiveness of NFPO in capturing the multi-modal action distributions and stabilizing learning.

**Weaknesses:**

- Considering this is the ICLR submission, the theoretical analysis may be more important than the engineering implementation and results. But the theoretical analysis for the algorithm and mathematical proofs in this paper are limited, e.g., one may expcet to see the analysis on the stability of NFPO and the reason why adding entropy loss in NFPO does not bring a significant performance difference.

- In some tasks like  MJP-PandaOpenCabinet and MJP-Go1JoystickRoughTerrain, NFPO fails to learn a good policy, which shows the limitation on the generalization ability of NFPO.

- Runtime overhead is reported but not decomposed. And there is no complexity analysis vs. the action dimension or number of coupling layers.

**Questions:**

- As stated in weaknesses, the theoretical analysis and mathematical proofs in this paper are limited. Can the authors provide more theoretical analysis on why NFPO can work better?

- Why tanh is more robust than clip in the authors' implementation?

- Can the authors provide more experiments with different hyperparameter combinations to show the robustness of NFPO?

---

> ### Author Response · Authors · 2025-11-21
>
> W1: *Considering this is the ICLR submission, the theoretical analysis may be more important than the engineering implementation and results...*
>
>
> R1:
> We understand it would be beneficial to have solid theoretical theory in this work, however:
>
> 1. to the best of our knowledge, there is no official announcement or guidance of ICLR stating that the theoretical analysis is more important than the engineering implementation and results (and we don't think this paper is an engineering effort, please see General Response 2). As researchers may have different research interests and expertise, we think it's  unfair to require all  papers to have  theoretical analysis. For why tanh is more stable than clip, we have provided an additional theoretical analysis in Appendix A.1.
>
> 2. similar work on NFs [1,2] also come with limited theoretical analysis . For normalizing flows, its closed-form calculation of log probability naturally plugs into multiple training pipelines.
>
> >[1] Zhai, Shuangfei, et al. "Normalizing Flows are Capable Generative Models." Forty-second International Conference on Machine Learning.
> >
> >[2] Ghugare, Raj, and Benjamin Eysenbach. "Normalizing Flows are Capable Models for RL." arXiv preprint arXiv:2505.23527 (2025).
>
> $~$
>
> for the analysis on the stability of NFPO, please refer to newly-added Appendix A.1
>
> for the entropy loss, $H(\pi) = E_\pi[- \log \pi]$, as PPO is an on-policy method, and NF has closed form $\log\pi$,  in our experiments we estimate it via monte-carlo method and optimize from on-policy data buffer. the experimental results show there is either no significant gains or performance degradation as shown in sect. 5.1.
>
> for an in-depth analysis over the dynamics between normalizing flows and its entropy optimization under on-policy framework, we hope to leave it as future works as in this paper we focus on the successful training of NF as policy parameterization.
>
> $~$
>
>
> W2: *In some tasks like MJP-PandaOpenCabinet and MJP-Go1JoystickRoughTerrain, NFPO fails to learn a good policy, which shows the limitation on the generalization ability of NFPO.*
>
>
> R2:
> 	Firstly, as noted in General Response 1, we use the default hyperparameters for all the methods evaluated and don't tune their results for a fair comparison. In this sense, it's common for some methods to show not-the-best result as online RL is a complex and continual process. Also, NFPO still achieves advantageous learning result on complex tasks like URG-g, URG-h1 and MJP-G1JoystickRoughTerrain and  better result than other multi-modal methods (Meow and FPO)
>
> $~$
>
> W3: *Runtime overhead is reported but not decomposed. And there is no complexity analysis vs. the action dimension or number of coupling layers.*
>
>
> R3:
> 	firstly, if we recall the normalizing flows' sampling or inference process, i.e.
>
>  $\log \pi(a) = \log q(f_\theta(a)) + \sum\nolimits_j \log \vert  \frac{df_{\theta_j}(a)}{da} \vert$
>
> It could be obversed that the computation complexity against the coupling layer is linear $O(L)$. And the computation complexity against action (input) dimension, under odd-even masking, the computation involves 2 parts: $t_\theta(x_{id})$ and $s_\theta(x_{id})$, both map from $D/2$ to $D/2$ where $x_{id}$ is unchanged part of input vector and $D$ is input dimension. So in total, the computation complexity is also linear against data dimension $O(D)$
>
> $~$
>
> Q1: *As stated in weaknesses, the theoretical analysis and mathematical proofs in this paper are limited. Can the authors provide more theoretical analysis on why NFPO can work better?*
>
> R4:
> 1. Gaussian distribution has known theoretical limitaions to model multi-modal distribution. If we consider a distribution that is a sum of 2 Gaussian: $p(x) = \frac{1}{2} \mathcal{N}(x; - \frac{\Delta}{2}, \sigma^2) +  \frac{1}{2} \mathcal{N}(x; \frac{\Delta}{2}, \sigma^2)$. Using a single Gaussian to model it (via minimizing KL divergence) would get the closed-form result: $q(x)=\mathcal{N}(x; \mu, s^2)$ where $\mu^* = 0$ and $s^2 = \sigma^2 + \Delta^2 / 4$
> where the learned gaussian  simply  puts its mean in the middle of the 2 distribution and enlarges the standard deviation to cover the 2 peaks.
> 2. For why tanh is more stable than clip, we have provided an additional theoretical analysis in Appendix A.1.
>
> $~$
>
> Q2: *Why tanh is more robust than clip in the authors' implementation?*
>
> R5: For why tanh is more stable than clip, we have provided an additional theoretical analysis in appendix A.1.
>
> $~$
>
> Q3: *Can the authors provide more experiments with different hyperparameter combinations to show the robustness of NFPO?*
>
> R6: we have conducted this experiment and present the results in Appendix A.3

---

### Author Response · Authors · 2025-11-21
**General Response 4 | revised manuscript and newly-added contents**

we have added the following additional contents in appendix for your reference

1. A.1 an theoretical analysis about why tanh performs better than clip
2. A.2 additional experiments on classic Mujoco environements
3. A.3 more ablation studies of NFPO on number of layers and hidden size
4. A.4 more tuning results of Meow on URG-g1 and URG-go2
5. A.9 detailed description of each environemnt we use in this work

for manuscript, we have modified several sentences and paragraphs to improve the writing, the text would be marked in blue color for better visualization


an additional uploaded folder contains our implementation of meow and the steps to check our modifications to the original implementation

---

> ### Author Response · Authors · 2025-11-27
>
> 2 additional appendix is added: A.5 (Stabilized Deployment) and A.6 (The mode of Normalizing Flows)  to further improve our manuscript.
>
> And *the detailed description of each environemnt we use in this work* is now at A.11

---

### Author Response · Authors · 2025-11-21
**General Response 3 | theoretical analysis:**

We have provided a theoretical analysis on why tanh would lead to better stabliity than clip in appendix A.1

as this paper targets on real-world-oriented robotic policy learning which is highly complex (the policy learned is actually an upper-level policy, whose output would  undergo a low-level PD controller to output torque that in turn drives the robot system), there is not much room  for us to derive rigorous theoretical analysis. Similar works that focus on theoretical analysis or system stability would have their own contribution and left as future works.

---

### Author Response · Authors · 2025-11-21
**General Response 2 |  Engineering Effort?:**

we don't think this work is an engineering implementation work because

1. we identified the reason why normalizing flow may fall short during on-policy policy optimization setting in Section 4. To our knowledge, this is **the first time** such training instability is reported and addressed. And it aligns with similar findings in recent work where adding noise to dequantize the training samples help improve the training in dataset-based settings ([1,2]);

2.  By effectively addressing the training instability, we could successfully train NFPO in various simulation environments without tuning (IsaacGym & IsaacLab & Mujoco playground) and transfer it to real-world robotic platform with very minimum tuning efforts, which is also not reported in alternative RL-based multi-modal policy learning works where careful tuning is needed as in fpo and meow.

In summary, NFPO is a **stabilized multi-modal RL method** that features little tuning effort,  advantageous performance in complex robotic learning environments , compared to existing rl-absed multi-modal policy learning works.

>[1] Zhai, Shuangfei, et al. "Normalizing Flows are Capable Generative Models." Forty-second International Conference on Machine Learning.
>
>[2] Ghugare, Raj, and Benjamin Eysenbach. "Normalizing Flows are Capable Models for RL." arXiv preprint arXiv:2505.23527 (2025).

---

### Author Response · Authors · 2025-11-21
**General Response 1 | hyperparameters**

For all the experiments conducted, we use the default hyperparameters (which is chosen either by the official maintainer (unitree rl gym and mujoco playground) or by the original authors (fpo and meow)) to guarantee fairness. For NFPO, we just replace the policy network (with aligned network size) from PPO and reuse the value net and all the other hyperparameters (learning rate, discount factor, ppo's training epoch and number of minibatch).

In this sense, the original ppo method in each environment is already heavily-tuned by each maintainer with guarantee performance (otherwise one cannot safely transfers it to real-world robots).

In complex locomotion based environment like URG-g1 and URG-h1 or MJP-G1JoystickRoughTerrain, NFPO obtains significant performance gains (in 10 seeds) over deployment-ready PPO. We think this showcases its modeling capacity in complex, high-dimensional controlling tasks which is also supported by our successful real-world deployment.

---

### Author Response · Authors · 2025-11-21

We thank all the reviewers'  precious time and constructive suggestions, below we provide several general responses for common questions and  detailed replies in each  thread.

---

### Meta-Review · Area_Chair_C7Vi · 2026-01-06

**Summary:**

__Theoretical concerns__
* I agree with the authors that ICLR does not prioritize theoretical contributions above empirical work (which was a concern raised by one of the reviewers). However, the presented submission strikes a difficult in-between balance between these. The theoretical results are not particularly compelling, but the empirical investigation is similarly not very broad (it is restricted to a relatively small set of tasks and small set of baselines, with none of these tasks being the "standard" set except for in the Appendix).

__Baselines and fairness of tuning__
* Several reviewers expressed concerns about the baselines and the fairness of hyperparameter tuning. The authors compellingly rebutted this concern, by pointing out that they use the same hyperparameters as PPO and just change the policy representation. They also provide more results on different hyperparameters for the Meow baseline in the Appendix.

__Comparison to offline RL methods__
* the paper argues for the value of online RL throughout (as being more memory and compute efficient), and uses this to justify the inclusion of baselines (for example, not including SAC or TD3), which was brought up as a concern by one reviewer. In the rebuttal period, the authors ran a single set of experiments comparing to SAC and TD3 in the Mujoco control suite, where SAC and TD3 outperform NFPO in every environment, sometimes by several multiplicative factors (in asymptotic performance). The paper also does not provide a comparison on compute or memory efficiency of NFPO to these stronger offline algorithms. As a result, it seems like NFPO would not be the algorithm of choice in many of these domains.

__Impact and similarity to related work__
* the work is very similar to several others that have used normalizing flows in RL, just not specifically the online setting. However, the challenges identified in the stability of using NFs in online RL are not specific to online RL, they seem to be an issue which applies any time you use RealNVP to model a policy. Since this is also the case in the offline RL setting, which has already been addressed extensively in related work (as highlighted by multiple reviewers), this distinction does not seem compelling.

For these reasons, I recommend rejection of the paper in its current form, and would encourage the authors to improve the comparisons to offline algorithms (including their asymptotic performance in the main paper, and comparing compute / memory efficiency to justify the online setting) in a future iteration.

**Reviewer Concerns:**

__Addressed concerns__
* hyperparameter tuning and fairness
* additional theoretical analyses of complexity

__Unaddressed concerns__
* inclusion of offline method comparisons
* sufficient distinction from related work in combining NFs with offline RL

**Reviewer Scores:**

I do not believe any reviewer would have changed their score, with the possible exception of reviewer AJ7p. However, while reviewer AJ7p had several concerns addressed, the experiments that were added to the appendix (which demonstrate that SAC and TD3 have better asymptotic performance than NFPO) would have likely caused them to maintain a score of reject. However, the discussion closed before they could respond to these updates.

---

### Decision · Program_Chairs · 2026-01-26

Reject